# Zika virus replicates in adult human brain tissue and impairs synapses and memory in mice

Claudia P. Figueiredo[1,9], Fernanda G.Q. Barros-Aragão [1,2,9], Rômulo L.S. Neris[3,9], Paula S. Frost [1,2], Carolina Soares[1,2], Isis N.O. Souza [1], Julianna D. Zeidler[4], Daniele C. Zamberlan [1], Virginia L. de Sousa[1], Amanda S. Souza [5], André Luis A. Guimarães [1], Maria Bellio [3], Jorge Marcondes de Souza[6], Soniza V. Alves-Leon[6,7], Gilda A. Neves [2], Heitor A. Paula-Neto[1], Newton G. Castro [2], Fernanda G. De Felice [4,8], Iranaia Assunção-Miranda [3], Julia R. Clarke[1], Andrea T. Da Poian [4] & Sergio T. Ferreira [4,5]

Neurological complications affecting the central nervous system have been reported in adult patients infected by Zika virus (ZIKV) but the underlying mechanisms remain unknown. Here, we report that ZIKV replicates in human and mouse adult brain tissue, targeting mature neurons. ZIKV preferentially targets memory-related brain regions, inhibits hippocampal long-term potentiation and induces memory impairment in adult mice. TNF-α upregulation, microgliosis and upregulation of complement system proteins, C1q and C3, are induced by ZIKV infection. Microglia are found to engulf hippocampal presynaptic terminals during acute infection. Neutralization of TNF-α signaling, blockage of microglial activation or of C1q/C3 prevent synapse and memory impairment in ZIKV-infected mice. Results suggest that ZIKV induces synapse and memory dysfunction via aberrant activation of TNF-α, microglia and complement. Our findings establish a mechanism by which ZIKV affects the adult brain, and point to the need of evaluating cognitive deficits as a potential comorbidity in ZIKV-infected adults.

[1] School of Pharmacy, Federal University of Rio de Janeiro, Rio de Janeiro, RJ 21944-590, Brazil. [2] Institute of Biomedical Sciences, Federal University of Rio de Janeiro, Rio de Janeiro, RJ 21944-590, Brazil. [3] Institute of Microbiology Paulo de Goes, Federal University of Rio de Janeiro, Rio de Janeiro, RJ 21944-590, Brazil. [4] Institute of Medical Biochemistry Leopoldo de Meis, Federal University of Rio de Janeiro, Rio de Janeiro, RJ 21944-590, Brazil. [5] Institute of Biophysics Carlos Chagas Filho, Federal University of Rio de Janeiro, Rio de Janeiro, RJ 21944-590, Brazil. [6] Division of Neurosurgery and Division of Neurology/Epilepsy Program, Clementino Fraga Filho University Hospital, Federal University of Rio de Janeiro, Rio de Janeiro, RJ 21944-590, Brazil. [7] Graduate Program in Neurology, Federal University of Rio de Janeiro State, Queen's University, Kingston, ON, Canada. [8] Centre for Neuroscience Studies & Department of Psychiatry, Queen's University, Kingston, ON, Canada. [9] These authors contributed equally: Claudia P. Figueiredo, Fernanda G. Q. Barros-Aragão, Rômulo L. S. Neris. Correspondence and requests for materials should be addressed to C.P.F. (email: claudia@pharma.ufrj.br) or to A.T.D.P. (email: dapoian@bioqmed.ufrj.br) or to S.T.F. (email: ferreira@bioqmed.ufrj.br)

Zika virus (ZIKV) is an arbovirus belonging to the *Flaviviridae* family responsible for a recent major outbreak in Latin America[1]. To date, several countries in Asia and in the Americas have reported active transmission of ZIKV[2]. Phylogenetic analysis of ZIKV strains isolated in the Americas revealed that they are closely related to Asian strains originally associated with neurological damage[5]. A causal relationship between intra-uterine viral infection and neonatal microcephaly has been established[4,6], and the mechanisms involved in developmental brain defects induced by ZIKV have been extensively studied[3,7].

In addition to complications arising from congenital infection, a number of reports have shown that adult individuals are susceptible to neurological complications following ZIKV infection[8–23]. These include Guillain-Barré syndrome[8,9,11,12,21], acute myelitis[8,13,21], encephalomyelitis[8,11,14,15,22], encephalitis[8,11,16,21], meningoencephalitis[18,19] and sensory polyneuropathy[20]. ZIKV has been detected in the brain and cerebrospinal fluid of adults with ZIKV-induced neurological disorders[8,10,12,18,22]. Moreover, a study conducted in Brazil showed increased incidence of serious neurological disorders in adult ZIKV-infected patients[8]. These findings indicate that ZIKV is highly neurotropic and reaches the mature central nervous system (CNS) following infection, implying that the virus can be harmful to the adult as well as to the developing brain[7].

In sharp contrast with a considerable body of knowledge now available regarding affected brain regions, cell types, and mechanisms of damage to the developing brain by ZIKV, much less is known on the susceptibility of the mature CNS to viral infection and on the consequences of infection in terms of behavioral and neurological manifestations. Using ex vivo human adult cortical tissue, we here show that ZIKV replicates in adult human brain tissue, infecting mature neurons. In adult immunocompetent mice infected by intracerebroventricular infusion of ZIKV, high levels of ZIKV RNA are detected in the frontal cortex and hippocampus, and mature neurons are the main cell type infected. Elevated brain levels of TNF-α, intense microgliosis, upregulated expression of complement system proteins (C1q and C3), and hippocampal synapse damage are verified in ZIKV-infected mice. Consistent with targeting of cognitive centers, infection impairs synapse function and memory in adult mice. Interestingly, neutralization of TNF-α or complement proteins, as well as blockage of microglial activation, rescues synapse/memory dysfunction in infected mice. These findings establish that ZIKV infection of the mature CNS causes synaptic dysfunction, and indicate that memory and cognition should be carefully evaluated in follow-up studies of adult patients infected by ZIKV.

## Results

**ZIKV replicates in adult human brain tissue.** Since ZIKV has been detected in the brains and CSF of adult patients[8,10,12,18,22], we first investigated whether ZIKV replicates in human brain tissue by exposing cultured adult temporal lobe cortical slices to the virus (Fig. 1a). ZIKV titer in the culture medium increased up to 48–72 h following washout (Fig. 1b), indicating successful viral replication and release of infectious particles from brain tissue. Active viral replication in human brain slices was further confirmed by detection of ZIKV polyprotein (NS2B) 24 h after infection (Fig. 1d, g, j). NS2B immunoreactivity colocalized with NeuN-immunoreactive cells (mature neuron marker) (Fig. 1c–e), but not with GFAP (astrocyte marker) (Fig. 1f–h) or F4/80 (microglia marker) (Fig. 1i–k), indicating that neurons are targeted by ZIKV in the mature human tissue. Controls showed lack of NS2B immunoreactivity in mock-infected human tissue as well as in the absence of primary anti-NS2B antibody (not shown).

**ZIKV infects neurons in the mature mouse brain.** To investigate the impact of infection on the mature CNS, we infused ZIKV into the lateral brain ventricle of adult mice (Fig. 2a), and viral replication was determined by qPCR (Fig. 2b). A progressive increase in viral RNA content was detected in the brains of ZIKV-infused mice, peaking at 6 days post-infection (dpi). ZIKV RNA was still detected at 15 and 30 dpi, followed by a progressive decrease until 60 dpi, when viral load reached the detection limit (Fig. 2b). Quantification of ZIKV-negative RNA in the brains of mice showed that replication of ZIKV was active at 6 dpi, and was undetectable at 30 and 60 dpi (Supplementary Fig 1a). While infection caused no mortality (Supplementary Fig. 1b), an important reduction in body weight was observed in infected mice compared to control groups (Fig. 2c).

We next asked whether ZIKV differentially targeted distinct CNS structures by examining viral load in frontal cortex, hippocampus, cerebellum, hypothalamus, striatum, and spinal cord 6 days after infection (Fig. 2d). Viral RNA levels were highest in brain regions involved in cognitive/memory processing (frontal cortex and hippocampus) and motor function (striatum) (Fig. 2d). While we found ~$10^5$ Eq PFU/mg ZIKV RNA in the frontal cortex of ZIKV-infected mice, other functional cortical areas, including sensory, entorhinal, and motor cortices, showed much lower viral loads ($10^3$–$10^4$ Eq PFU/mg) (Supplementary Fig. 1c).

To examine the possibility that viral distribution in specific brain structures was a consequence of their proximity to the site of viral infusion, another group of animals received ZIKV infusion into the third brain ventricle. Similar to mice infused into the lateral ventricle, mice infused into the third ventricle exhibited highest ZIKV RNA loads in the frontal cortex, hippocampus, and striatum (Supplementary Fig. 1d), suggesting that the virus preferentially targets these structures independently of the site of injection into the CNS.

To examine the possibility that the regional specificity of viral distribution in the rodent brain was due to a large viral input, we infected mice with $10^3$ PFU ZIKV, i.e., 100-fold lower than in previous experiments. Similar to animals infected with $10^5$ PFU, ZIKV targeted preferentially frontal cortex, hippocampus, and striatum (Supplementary Fig. 1f) of mice infected with $10^3$ PFU. This suggests that these brain regions are preferentially targeted regardless of the viral amount reaching the CNS.

We also noted that, following infusion into either the lateral (Fig. 2e) or third (Supplementary Fig. 1e) brain ventricles, ZIKV RNA was detected, albeit at lower levels than in the CNS, in sciatic nerve, dorsal root ganglion, spleen, testes, and liver. This suggests that ZIKV escapes the CNS and reaches peripheral tissues after replication in the brain.

Previous studies have shown that ZIKV fails to infect immunocompetent mice when administered through a peripheral route[24], leading to the use of immunocompromised mice to investigate the neurological impact of infection[25]. Interestingly, we found that peripheral infection by ZIKV in interferon receptor 1 knockout adult mice (A129) or in wild-type neonatal mice (a stage when interferon-mediated inflammatory response is inefficient) resulted in high and similar levels of viral RNA in all brain regions analyzed (Supplementary Fig. 1g, h). These results suggest that the susceptibilities of distinct brain structures to infection by ZIKV are likely related to their differential abilities to mount an effective interferon-based immunological response. We further found that ZIKV reaches the frontal cortex and hippocampus of wild-type adult Swiss mice following peripheral administration, but does not replicate effectively when administrated by this route (Supplementary Fig. 1i).

Immunostaining for ZIKV NS2B polyprotein in brain sections from mice infected by i.c.v. route (Fig. 2f) revealed robust staining

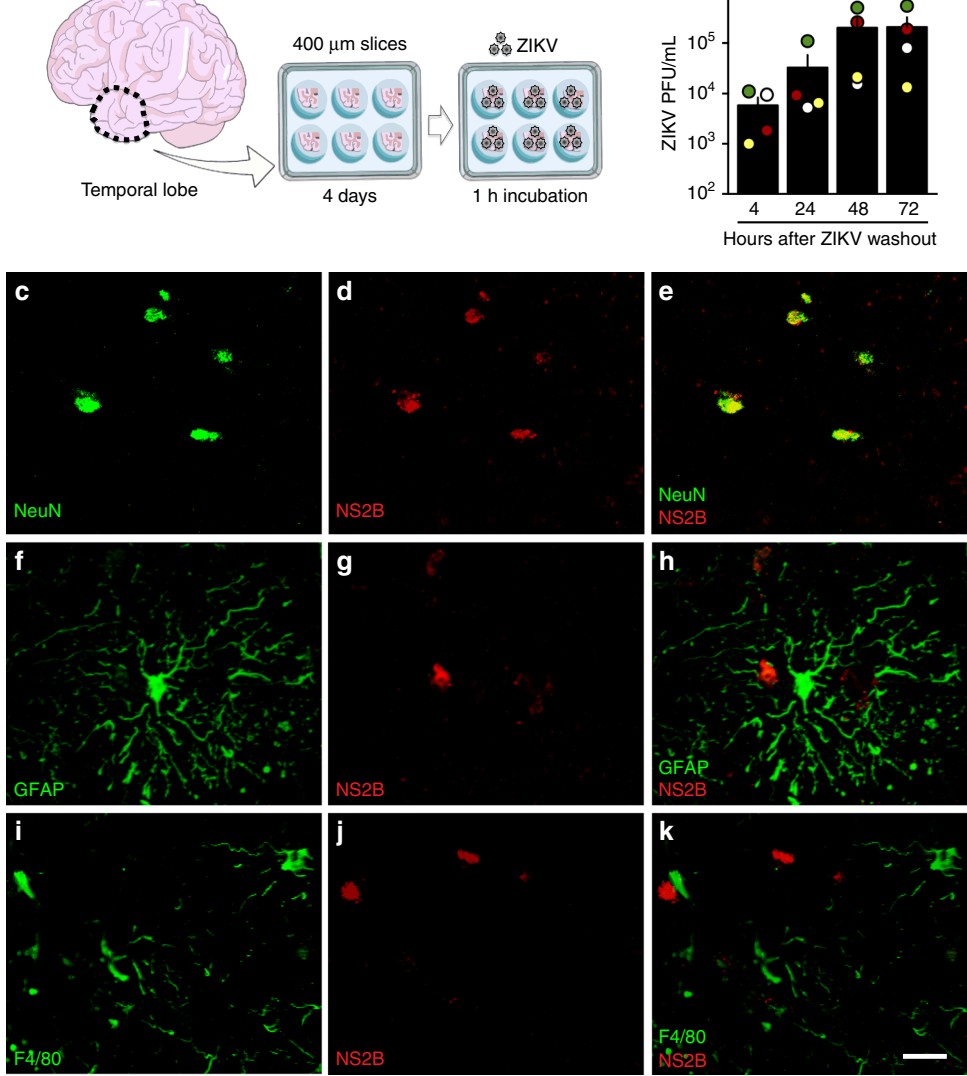

**Fig. 1** ZIKV replicates and targets neurons in adult human cortical tissue. **a** Human temporal lobe cortical tissue fragments were sectioned into 400 μm slices and maintained in culture for 4 days prior to incubation with ZIKV ($10^7$ PFU) or mock medium for 1 h, followed by washout. Slices were used for immunohistochemistry, and culture medium was collected for determination of virus titers. **b** Quantitation of infectious viral particles in the culture medium by plaque assay. Bars represent means ± SEM. Symbols represent means of duplicate or triplicate determinations from four independent human donors (shown in different colors) at each time point. (*$p = 0.0411$, input (4 h after washout) vs 48 or 72 h after washout; Friedman followed by Dunn's post-test). **c–k** Representative images of double immunofluorescence labeling for ZIKV (NS2B protein, red) and NeuN (green) (**c–e**), GFAP (f–h), or F4/80 (**i–k**) in ZIKV -infected human tissue slices from two independent donors. Scale bar = 50 μm. Source data from panel **b** are provided as Source Data File

in the hippocampal pyramidal cell layer (Fig. 2g–i) and in the granular layer of the dentate gyrus (Fig. 2g, j). Some ZIKV-infected mice showed areas of necrosis next to viral immunor-eactivity (Fig. 2i, white arrows). We also found abundant NS2B-positive cells in the striatum (Fig. 2k, l) and frontal cortex (Fig. 2m), but not in the thalamus (Th) (Fig. 2n) or sub-ventricular zone (Fig. 2o). In agreement with findings in adult human tissue, NS2B immunostaining colocalized with NeuN-positive cells (Fig. 2p–r), and no colocalization was observed between NS2B immunoreactivity and either GFAP (Fig. 2s–u) or F4/80 (Fig. 2v–x). Controls showed lack of NS2B immunoreactivity in the brains of mock-infused mice, as well as in the absence of primary NS2B antibody. Results indicate that ZIKV targets memory-related brain regions, and neurons are the main target of infection in immunocompetent adult mice.

Because neural progenitor cells (NPCs) have been shown to be major targets of ZIKV infection in interferon-deficient mice[25], we

further investigated whether such cells were targeted in adult immunocompetent mice. No colocalization between NS2B and doublecortin (DCX, a NPC/immature neuron marker) immu-nostaining was detected in the hippocampal dentate gyrus of infected mice (Supplementary Fig. 2a–c). However, we found that the number of DCX-positive cells in the dentate gyrus was reduced in ZIKV-infected mice (Supplementary Fig. 2d–f).

**ZIKV causes synapse and memory dysfunction in adult mice.** Our finding that ZIKV-targeted memory-related brain regions prompted us to examine the impact of infection on memory. As expected, mock-infused mice learned the novel object recognition (NOR) memory task, as demonstrated by longer exploration of the novel object over the familiar one (Fig. 3a–d, white bars). In contrast, ZIKV-infected mice failed the NOR task as early as 1 dpi, and memory impairment persisted at 14 and 30 dpi (Fig. 3a–c, black bars). ZIKV-infected mice further showed memory

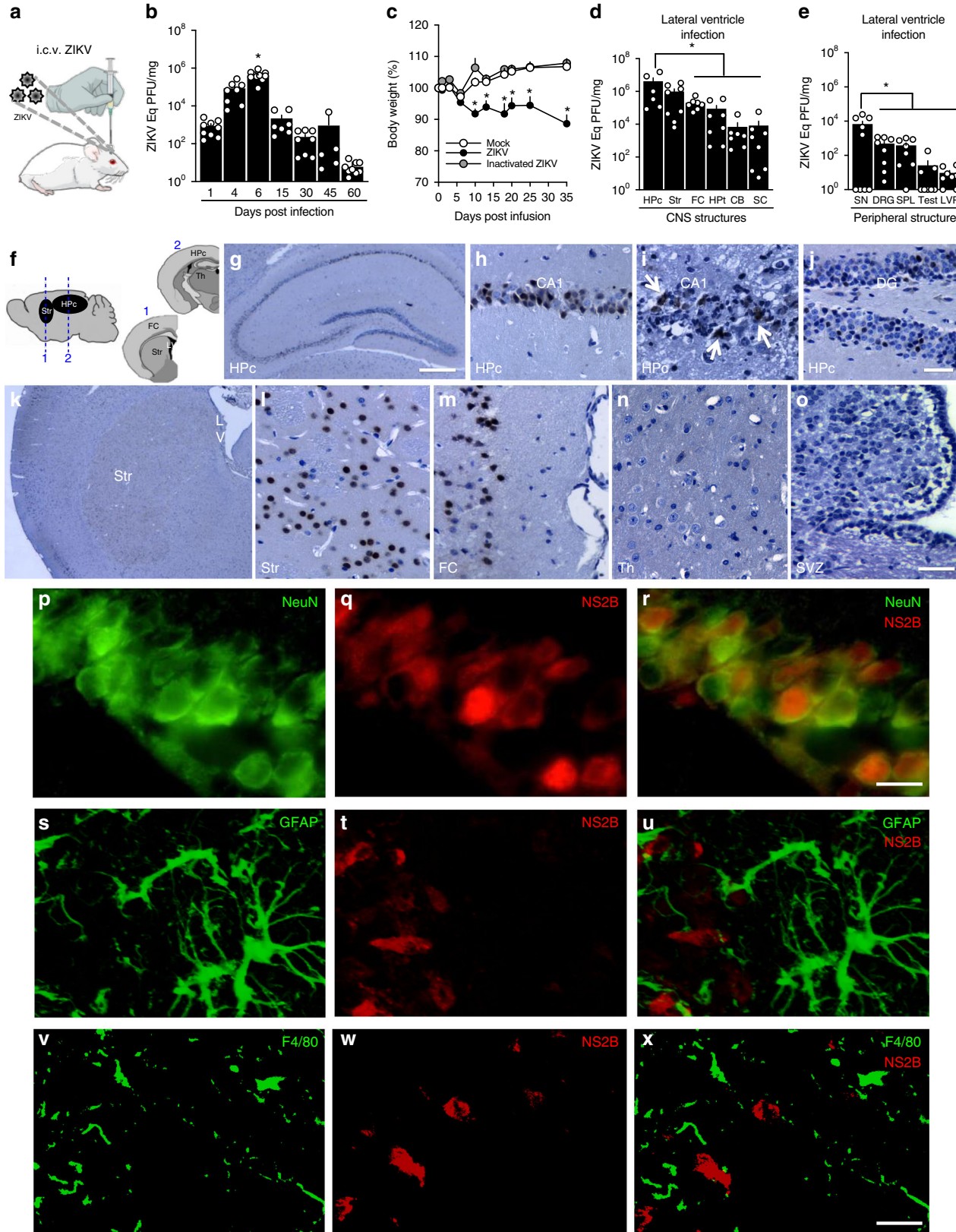

impairment in a hippocampus-dependent task, the passive avoidance (step-down) paradigm (Fig. 3e, Supplementary Fig. 4i). Interestingly, performances of infected mice in both NOR (Fig. 3d) and step-down (Fig. 3f; Supplementary Fig. 4j) tests returned to normal at 60 dpi, when ZIKV RNA approached non-detectable levels in the brains of infected mice (Fig. 2b).

To determine if presence of viral protein alone was sufficient to induce memory impairment, mice were infused with $10^5$ PFU UV-inactivated ZIKV (iZIKV) and evaluated in the NOR task. iZIKV did not impact memory function in mice (Supplementary Fig. 4k–o), suggesting that memory impairment requires active ZIKV replication. To further investigate if impairment triggered

**Fig. 2** ZIKV replicates and targets memory-related areas in adult mouse brain. **a** Adult mice received an infusion of $10^5$ PFU ZIKV, UV-inactivated ZIKV or mock medium into the cerebral lateral ventricle. **b** At the indicated days post-infection (dpi), brains were processed for determination of ZIKV RNA levels by qPCR (F(6,49) = 25.22; *$p < 0.0001$ comparing 6 dpi to other time points, one-way ANOVA followed by Bonferroni; $N = 10$ (6 dpi); 9 (1 and 60 dpi); 8 (4 and 30dpi); 7 (15dpi) and 5 (45dpi). **c** Body weight measured up to 35 dpi (F(2,27) = 18.11; *$p < 0.0001$ compared to mock-infused mice, repeated measures ANOVA followed by Bonferroni; $N = 10$ mice/group; graph represents one of two independent experiments). **d**, **e** Distinct CNS (**d**) or peripheral (**e**) structures were isolated at 6dpi and processed for determination of ZIKV RNA levels by qPCR. (In **d** F(4,38) = 3.096; * indicates: $p = 0.0205$ Hpc vs FC, $p = 0.0125$ Hpc vs Hpt, $p = 0.0137$ Hpc vs CB, $p = 0.0105$ Hpc vs SC, one-way ANOVA followed by Bonferroni. In **e**: F(4,36) = 4.014; *$p = 0.0206$ SN vs DRG, $p = 0.0181$ SN vs SPL, $p = 0.0112$ SN vs LVR, $p = 0.0115$ SN vs Testes, one-way ANOVA followed by Bonferroni; $N = 6$ (Hpc hippocampus); 9 (SN sciatic nerve); 8 (Str striatum, Hpt hypothalamus, SC spinal cord, DRG dorsal root ganglion, SPL spleen, LVR liver and Testes); 7 (FC and CB). Bars represent means ± SEM. Symbols represent individual mice. **f** At 6 dpi, brains were processed for immunohistochemistry. (**g-o**) Representative ZIKV immunolabeling (NS2B protein) in distinct brain regions from ZIKV-infected mice: hippocampus (**g**), CA1 hippocampal subregion (**h**, **i**), dentate gyrus (DG) (**j**), striatum (**k**, **l**), frontal cortex (**m**), thalamus (Th) (**n**), sub-ventricular zone (SVZ) (**o**). LV lateral ventricle. Arrows point to areas of necrosis. **p-x** Representative double immunofluorescence images for ZIKV (NS2B protein, red), NeuN (**p-r**), GFAP (**s-u**), or F4/80 (**v-x**) (green) in the CA1 hippocampal region of ZIKV-infected mice ($N = 4$). Scale bars = 200 μm (**g**, **k**), 40 μm (**h-j**), 50 μm (**l-o**), 10 μm (**p-r**), 15 μm (**s-x**). Source data from panels **b-e** are provided as Source Data File

by ZIKV could be a non-specific response triggered by replication of any arbovirus, mice were infected i.c.v. with $10^5$ PFU of Mayaro virus (MAYV). Results showed that MAYV infection had no impact on memory at any time point evaluated (Supplementary Fig. 4p–u).

Viral infections may cause non-specific behavioral symptoms known as sickness behavior[26]. To examine the possibility that ZIKV infection caused sickness behavior that could affect memory function, mice were assessed in a number of tests to evaluate locomotor/exploratory activities as well as anxious- and depressive-like behaviors. Mock-infused and ZIKV-infected mice showed similar exploratory behavior when sequentially evaluated in an open-field arena from 1 to 60 dpi (Supplementary Fig. 3a, f). At longer evaluation times in the same task during acute infection, both experimental groups showed the expected habituation behavior (Supplementary Fig. 3b) as well as similar distances travelled, numbers of body rotations, and rearings (Supplementary Fig. 3c–e). Further, both groups explored similarly the center of the open-field arena and the open arms of the elevated plus maze (Supplementary Fig. 3f–h). Collectively, results indicate that ZIKV infection did not affect locomotor/ exploratory activities or anxiety in mice.

We further found no differences between mock-infused and ZIKV-infected animals in immobility time in the tail suspension test (TST) (Supplementary Fig. 3i) or in grooming behavior in the sucrose splash test (SST) (Supplementary Fig. 3j, k), tasks designed to evaluate behavioral depressive-like[27] and anhedonic[28] behavior, respectively. Additional control measurements showed that neither groups of mice had innate preferences for the objects used in the NOR memory test (Supplementary Fig. 4a–d), and that infection did not affect the total object exploration time in the test (Supplementary Fig. 4e–h). Altogether, results indicate that ZIKV infection did not instigate depressive-like/sickness behavior in mice.

Synapse damage is a common denominator in a number of memory-affecting conditions[29]. We thus investigated whether infection by ZIKV-induced synapse damage. Decreased colocalization between synaptophysin (SYP, a pre-synaptic marker) and Homer-1 (a post-synaptic marker) immunoreactive puncta, a measure of synaptic density, was observed in hippocampal CA3 region (Fig. 3g–k) and dentate gyrus (DG) (Supplementary Fig. 5a–e) of ZIKV-infected mice at 6 dpi, indicating that infection led to synapse damage and elimination. No changes in levels of SYP or PSD-95 (a post-synaptic marker) were detected by immunoblotting in total hippocampal homogenates from ZIKV-infected mice (Supplementary Fig. 5f, g). In agreement with the temporal recovery in memory performance, SYP-positive and colocalized SYP/Homer-1 synaptic puncta returned to control levels in the

hippocampal CA3 region of ZIKV-infected mice at 60 dpi (Fig. 3l–p). No changes in SYP, Homer-1 or colocalized SYP/Homer-1 synaptic puncta were observed in hippocampal CA1 region (Supplementary Fig. 5h–l), frontal cortex (Supplementary Fig. 5m–q), or striatum (Supplementary Fig. 5r–v) of ZIKV-infected mice at 6 dpi.

We next investigated whether synapse damage/elimination induced by ZIKV infection in adult mice was associated with impaired synaptic function. Because the CA3 hippocampal region was found to be particularly vulnerable to synapse damage (Fig. 3g–k), we induced long-term potentiation (LTP) at DG-CA3 synapses in the mossy fiber pathway in hippocampal slices from mock or ZIKV-infected mice. In slices from mock-infused mice, evoking two excitatory post-synaptic field potentials (fEPSPs) 40 ms apart revealed marked paired-pulse facilitation, and high-frequency stimulation induced robust post-tetanic potentiation (PTP) ($400 \pm 45\%$ of baseline) that slowly decayed to a persistently potentiated level ($137 \pm 8\%$, 35–40 mins following stimulation) (Fig. 3q–t). In contrast, both PTP and long-term potentiation (LTP), as well as paired-pulse facilitation, were reduced in slices from ZIKV-infected mice, indicating impaired synaptic function.

Phosphorylation of cAMP-responsive element binding protein at serine residue 133 (CREBpSer$^{133}$) plays a key role in hippocampal memory consolidation[30]. Consistent with impaired memory, we further found a marked decrease in CREBpSer$^{133}$-immunoreactivity in hippocampus (Supplementary Fig. 6a–f) and parietal cortex (Supplementary Fig. 6g–i) of ZIKV-infected mice at 6 dpi.

To determine whether synapse damage and inhibition of synaptic plasticity induced by ZIKV were accompanied by neurodegeneration, FluoroJade B (FJ) staining of brain sections was performed at 6 and 10 dpi. No FJ-positive cells were found in hippocampal CA1, CA2, CA3, dentate gyrus or parietal cortex of ZIKV-infected mice at 6 dpi (Supplementary Fig. 7a–c), and few FJ-positive cells were seen in CA1 (but not dentate gyrus, CA3 or parietal cortex) in two out of five mice at 10 dpi (Supplementary Fig. 7d–f).

**ZIKV triggers brain inflammation in adult mice.** Inflammation is an important feature of brain viral infections, and has been associated with memory impairment in pathological conditions[31–33]. Brain expression of the proinflammatory cytokines IL-1β (Fig. 4a) and, notably, TNF-α (Fig. 4b) was significantly elevated at 6 dpi (black bars), and returned to near control levels 30 dpi (Fig. 4a, b). Expression of these cytokines was unaffected in the brains of mice infused with iZIKV (gray bars), indicating that the proinflammatory response required active viral replication.

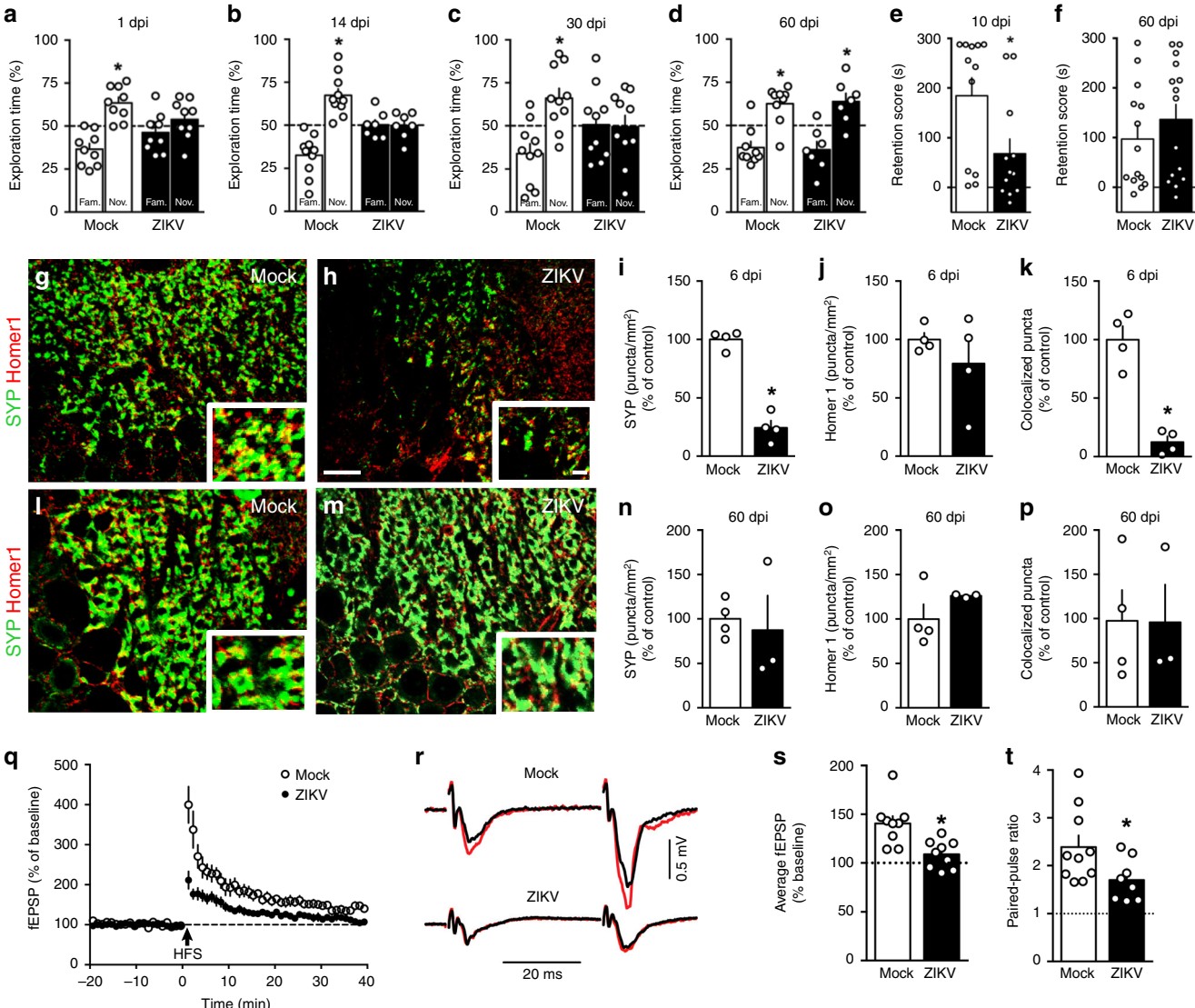

**Fig. 3** ZIKV causes synapse damage and memory impairment in mice. Adult mice received an i.c.v. infusion of $10^5$ PFU ZIKV or mock medium. **a–d** Independent groups of mice were tested in the novel object recognition (NOR) test at 1 dpi (**a** t = 4.553; *p = 0.0014; N = 10 mice/group), 14 dpi (**b** t = 4.587; *p = 0.0013; N = 10 (mock), and 7 (ZIKV)), 30 dpi (**c** t = 2.714; *p = 0.0239; N = 10 mice/group), or 60 dpi (**d** t = 3.525; *p = 0.0078 for mock-infused; t = 2.907; p = 0.0271 for ZIKV-infected mice; N = 9 (mock) and 7 (ZIKV)); one-sample Student's t-test compared to fixed value of 50%. **e, f** Mice were tested in the step-down inhibitory avoidance test at 10 dpi (**e** t = 2.476; *p = 0.0215; Student's t-test; N = 12 mice/group) or 60 dpi (**f** N = 14 (mock) and 15 (ZIKV)). **a, b, c, e** represent one of two independent experiments. Representative images of the CA3 hippocampal region of mock-infused (**g, l**) or ZIKV-infected mice (**h, m**) at 6 (**g, h**) and 60 (**l, m**) dpi, immunolabeled for synaptophysin (SYP, green) and Homer1 (red). (**i–k, n–p**) Number of puncta for SYP (**i**), Homer1 (**j**), and colocalized SYP/PSD-95 puncta (**k**). (**i** t = 10.46, *p < 0.0001; **k** t = 6.009, p = 0.0004, Student's t-test; N = 4 mice/group; **n–p** N = 4 (mock) and 3 (ZIKV)). Scale bar = 25 μm; inset scale bar = 5 μm. **q–t** ZIKV infection impairs LTP induction in dorsal hippocampal CA3 area. **q** Plot shows means ± SEM of the slopes of field excitatory post-synaptic potentials (fEPSPs) evoked in CA3 by mossy fiber stimulation, before and after high-frequency stimulation (HFS), in acute slices from mock- or ZIKV-infected mice. **r** Representative fEPSPs recorded before (black traces) and 35–40 min after (red traces) HFS. **s** Average fEPSP slopes 35–40 min post-HFS. **t** Baseline paired-pulse amplitude ratios. (N = 10 slices from 8 mock-infused and 8 slices from 5 ZIKV-infected mice; s: t = 2.733, *p = 0.015; t: t = 2.228 *p = 0.041; Student's t-test). Bars represent means ± SEM. Symbols represent individual mice (**a–p**) or individual slices (**s–t**). Source data from panels **a–f**, **i–k**, **n–t** are provided as Source Data File

We next investigated whether gliosis was induced by infection. Mouse brain sections obtained 6 dpi were immunolabeled for GFAP, ionized calcium binding adaptor molecule 1 (Iba-1, a macrophage/microglial marker) and transmembrane protein 119 (TMEM119, a microglial marker). No differences in GFAP immunoreactivity were detected in the hippocampus (Supplementary Fig. 8a–c) or parietal cortex (Supplementary Fig. 8d–f) of ZIKV-infected mice compared to controls. In contrast, intense Iba-1 immunoreactivity was verified in the brains of ZIKV-infected mice at 6 dpi, notably in the hippocampus (Fig. 4c, f–j).

Hippocampi of ZIKV-infected mice exhibited increased numbers of Iba-1-positive cells (Fig. 4d, f–j) with increased Feret's diameters (Fig. 4e), suggesting a predominantly amoeboid cell morphology typical of an activated phagocytic state (Fig. 4I, j). Perivascular cuffings and microglial nodules (Fig. 4j), characteristic of viral encephalitis, were also found in ZIKV-infected brains. Iba-1 immunostaining in hippocampi of ZIKV-infected mice was reduced by 30 dpi and reached control levels at 60 dpi (Fig. 4c). Further indicating that microgliosis was induced by infection, we found significantly higher TMEM119

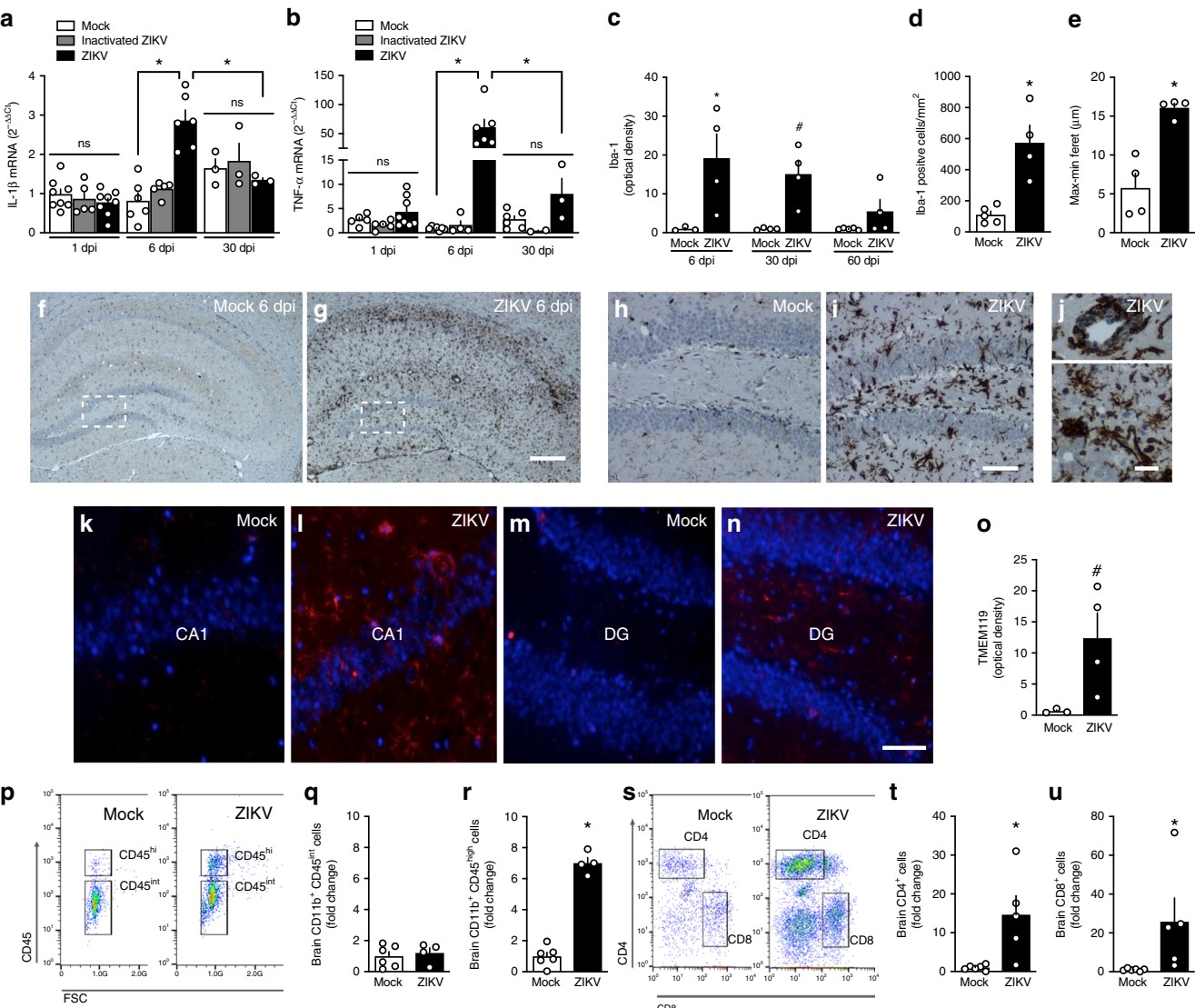

**Fig. 4** ZIKV triggers brain inflammation and microgliosis in adult mice. Adult mice received an i.c.v. infusion of $10^5$ PFU ZIKV, UV-inactivated ZIKV or mock medium. **a** Brains were processed for determination of the expression of IL-1β (**a** $F_{(2,38)} = 4.878$, *$p < 0.0001$, ZIKV-infected vs mock-infused mice at 6dpi, two-way ANOVA followed by Bonferroni; $N = 8$ (1 dpi), 6 (6 dpi), and 3 (30 dpi) (mock); 5 (1 and 6 dpi) and 3 (60 dpi) (inactivated-ZIKV); and 8 (1 dpi), 6 (6 dpi) and 3 (30 dpi) (ZIKV)mice) or TNF-α (**b** $F_{(2,37)} = 13.62$, *$p < 0.0001$, ZIKV-infected vs mock-infused mice at 6 dpi; two-way ANOVA followed by Bonferroni; $N = 5$ (1 dpi), 7 (6 dpi) and 6 (30 dpi)(mock); 5 (1 dpi), 4 (6 dpi), and 2 (60 dpi) (inactivated-ZIKV) and 8 (1 dpi), 6 (6 dpi) and 3 (60 dpi) (ZIKV)mice. **c** Integrated Iba-1 immunoreactivities (optical density) in the hippocampi of ZIKV-infected mice at 6, 30, or 60 dpi ($F_{(2,10)} = 0.2612$, 6 dpi: *$p = 0.0110$, 30 dpi: #$p = 0.0554$; two-way ANOVA followed by Bonferroni compared to mock-infused mice at the same time point; $N = 3$ (mock) and 4 (ZIKV) mice at 7 dpi; 4 mice/group at 30 dpi and 5 (mock) and 4 (ZIKV) mice at 60 dpi. **d** Iba-1 positive cells in the hippocampi of mice at 6 dpi (t = 4.507; *$p = 0.0028$; Student's *t*-test; $N = 5$ (mock)and 4 (ZIKV) mice. **e** Maximum-minimum Feret index in microglia in hippocampi of mock-infused or ZIKV-infected mice at 6dpi (t = 5.205; *$p = 0.002$; $N = 4$ mice/group). **f, g** Representative images of Iba-1 immunoreactivity in the hippocampi of mock-infused (**f**) or ZIKV-infected (**g**) mice at 6dpi. Scale bar = 300 μm. **h, i** Higher magnification images of the regions defined by dashed white rectangles in panels **f** and **g**, respectively. Scale bar = 50 μm. **j** Representative images illustrating perivascular cuffings (top) and microglial nodules (bottom) in hippocampi of ZIKV-infused mice at 6 dpi. Scale bar = 20 μm. **k–n** Representative images of TEMEM119 immunolabeling in hippocampal CA1 region (**k, l**) and in DG (**m, n**) of mock-infused or ZIKV-infected mice at 6dpi. Scale bar = 50 μm. **o** Integrated TMEM119 immunoreactivities (optical densities) in the hippocampi of mice at 6 dpi (t(1,5) = 2.435; #$p = 0.059$, Student's *t*-test); $N = 3$ (mock) and 4 (ZIKV) mice. Bars represent means ± SEM. Symbols correspond to individual mice. **p–r** Representative dot plot (**p**) and frequency of quiescent microglia (CD11b+CD45int cells; **q**, and activated myeloid cells (CD11b+CD45high cells; **r** in brains of mock- or ZIKV-infused mice at 6 dpi (**q** $N = 6$ (mock) and 4 (ZIKV) mice; **r** t = 14.03; *$p < 0.0001$; Student's *t*-test; $N = 6$ (mock) and 4 (ZIKV) mice. **s–u** Representative dot plot (**s**) and frequencies of CD4 (**t**) and CD8-positive (**u**) cells in brains of mock- or ZIKV-infused mice at 6 dpi (**t** t = 3.072; *$p = 0.0133$; **u** t = 2.242; #$p = 0.0517$; $N = 6$ (mock) and 5 (ZIKV) mice; Student's *t*-test). Bars represent means ± SEM; Symbols represent individual mice (**a–o**) or a pool of two mice (**p–u**). Source data from panels **a–e**, **o, q, r, t, u** are provided as Source Data File

immunoreactivity in the CA1 and DG hippocampal sub-regions of ZIKV-infected mice (Fig. 4k–o).

Prominent microgliosis was also found in parietal and frontal cortices (Supplementary Fig. 9) and striatum (Supplementary Fig. 10a, b, g) at 6 dpi, consistent with presence of viral RNA in these brain regions at 6 dpi (Fig. 2d). We thus asked whether striatal viral load and neuroinflammation were associated with impaired performance in the rotarod test, a classical paradigm for

assessment of motor function. ZIKV-infected mice (5 dpi) exhibited lower latencies to fall from the rotating rod than mock-infected mice (Supplementary Fig. 10h). ZIKV-infected mice regained normal motor function at 30 dpi (Supplementary Fig. 10i), which was accompanied by return of striatal Iba-1 immunolabeling to control levels by 30 and 60 dpi (Supplementary Fig. 10c–g). These findings indicate that ZIKV-induced motor dysfunction is reversible and paralleled by microgliosis, similarly to memory impairment. In agreement with lower viral levels in sensory cortex compared to other cortical areas (Fig. 2d, Supplementary Fig. 1c), we found no significant impairment in peripheral sensitivity in ZIKV-infected mice compared to mock-infected animals (Supplementary Fig. 10j–l).

Further characterization of the cellular response to infection was performed by flow cytometry in brain cell suspensions from mock- and ZIKV-infected mice at 6 dpi. We found no differences in the number of $CD11b^+CD45^{int}$ cells (resting microglia) between groups (Fig. 4p, q; Supplementary Fig. 11a). On the other hand, ZIKV infection led to increased brain numbers of $CD11b^+CD45^{high}$ cells (Fig. 4p, r), which comprise a mixed population of infiltrating macrophages and activated microglia[34]. In addition, infiltrates of CD4+ and CD8+ lymphocytes were detected in the brains of infected mice (Fig. 4s–u, Supplementary Fig. 11b), indicating that both helper and cytotoxic lymphocytes, respectively, infiltrate the mouse brain following ZIKV infection.

**TNF-α drives microgliosis and synapse/memory deficits.** Pruning of synaptic terminals by microglia is an important mechanism underlying synapse loss in neurodegenerative disorders and in viral infections[31,33]. In line with this, and given the phagocytic appearance of microglia in ZIKV-infected mice, we hypothesized that microglial activation could play a central role in ZIKV-induced synapse damage and memory impairment. Three-dimensional image reconstructions of Iba-1-positive cells in the hippocampi of ZIKV-infected mice showed synaptophysin-positive terminals inside Iba-1-positive cells (Fig. 5a, b; Supplementary Video 1a, b), and revealed an increased number of pre-synaptic terminals inside Iba-1-positive cells compared to controls (Fig. 5c).

Next, we evaluated whether blockage of microglial activation could prevent ZIKV-induced memory impairment. For this, we used minocycline, a tetracycline antibiotic that blocks microglial activation[35]. Minocycline had no effect on the number of hippocampal Iba-1 positive cells (Fig. 5d–g) but, as expected, blocked the morphological conversion of hippocampal microglia from a more ramified (surveilling) to a more amoeboid (phagocytic) morphology (as indicated by Feret diameter analysis; Fig. 5d–f, h). Interestingly, minocycline treatment prevented ZIKV-induced memory impairment (Fig. 5i). Control measurements showed that minocycline had no effect on motivation or anxiety-like behaviors in mice (Supplementary Fig. 12a–d), and did not affect brain ZIKV replication (Supplementary Fig. 12e). We further found that minocycline prevented the inhibition of mossy fiber PTP, LTP and paired-pulse facilitation induced by ZIKV infection (Fig. 5j–m).

Intriguingly, we found that minocycline treatment did not block the increase in brain TNF-α expression (Fig. 5n) induced by infection. Given the known effect of TNF-α on microglial activation[36], we next treated ZIKV-infected mice with infliximab (a monoclonal TNF-α neutralizing antibody) and assessed hippocampal microgliosis and memory/synaptic functions. We found that treatment with infliximab attenuated ZIKV-induced hippocampal microgliosis (Fig. 6a–e) and the decrease in synaptic puncta (Fig. 6f–k). Importantly, infliximab restored memory (Fig. 6l) and synaptic (Fig. 6m–p) functions in ZIKV-infected mice.

Treatment with infliximab had no effect on total object exploration times, anxiety behavior (Supplementary Fig. 12f–i), or on ZIKV replication in the mouse brain (Supplementary Fig. 12j).

Because IL-1β levels were also elevated in the brains of ZIKV-infected mice (Fig. 4a), we examined the possibility that IL-1β could mediate memory deficits induced by infection. Contrary to this hypothesis, we found that mice lacking the IL-1β receptor ($Il1r^{-/-}$) were equally susceptible to ZIKV-induced memory impairment as wild-type mice (Supplementary Fig. 12k). Control experiments showed that deletion of IL-1β receptor had no effect on locomotion or exploration (Supplementary Fig. 12l–o).

Collectively, these findings suggest that microglia/macrophages, while ultimately responsible for hippocampal synapse damage, are not the major brain source of TNF-α in ZIKV-infected mice. TNF-α production appears to be upstream of microgliosis in the context of ZIKV infection, and is insufficient to cause cognitive impairment in the absence of microglial activation.

**Role of complement system proteins C1q and C3.** Complement system proteins C1q and C3 have been implicated in synaptic tagging driving microglial recognition and pruning of synapses in the brains of mice infected by West Nile virus (WNV)[31]. We thus measured the expression of C1q and C3 in the brains of mice following ZIKV infection. Both C1q (Fig. 7a) and C3 (Fig. 7b) were significantly upregulated 6 dpi, and returned to control levels 60 dpi. Blockage of microglial activation by minocycline attenuated upregulation of both C3 and C1q (Fig. 7c, d), while central (Fig. 7e, f) or peripheral (Supplementary Fig. 13a, b) inhibition of TNF-α-mediated signaling by infliximab attenuated ZIKV-induced upregulation of C3 but not C1q.

Soluble C1q and C3 proteins in the brain diffuse and target dysfunctional synapses, signaling for microglial pruning[31,33]. This prompted us to investigate whether blockage of soluble C1q and C3 by i.c.v. infusion of specific antibodies affected ZIKV-induced cognitive impairment. Control experiments showed that neutralizing each of these proteins in the brain had no effect on locomotion or anxiety behavior (Supplementary Fig. 13c–f). Notably, neutralization of either C1q or C3 rescued both hippocampal synapse density (Fig. 7g–m) and NOR performance (Fig. 7n) in ZIKV-infected mice.

Altogether, results suggest that ZIKV infection leads to elevated brain levels of TNF-α, microglial activation and increased expression of C1q and C3, culminating with synapse damage and memory impairment in adult mice.

**Discussion**
ZIKV infection during pregnancy causes serious neurological conditions in newborns, including microcephaly[4,6]. Though initially considered a self-limited febrile state in adults, ZIKV infection was recently linked to the development of neurological complications[8–23], including acute myelitis[13,21], encephalitis,[8,10,11,16,21] and meningoencephalitis[18,19]. Indeed, ZIKV infection has been associated with increased incidence of a broad spectrum of neurological disorders in adult patients[8]. Moreover, ZIKV has been found in the brains and CSF of adult patients[8,10,11,18,22], and recent studies demonstrated that ZIKV persists in CSF and lymph nodes of infected monkeys for several weeks after clearance from peripheral blood, urine, and mucosal secretions[37]. Collectively, these findings suggest that, in addition to the developing nervous system, the mature CNS is targeted by ZIKV. Nonetheless, while several lines of evidence indicate that human neural precursor cells are affected by ZIKV[3], whether

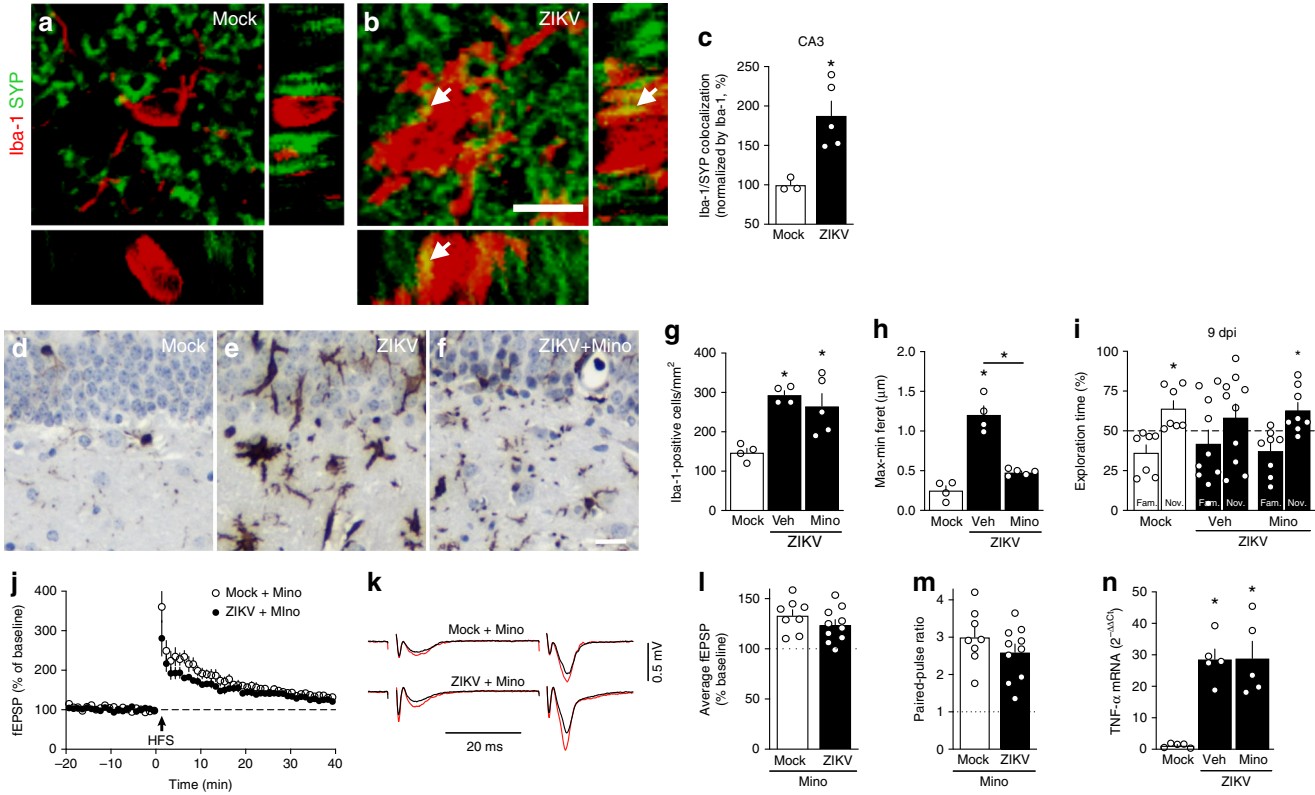

**Fig. 5** Microglia mediates ZIKV-induced memory impairment in mice. Adult mice received an i.c.v. infusion of $10^5$ PFU ZIKV or mock medium. **a**, **b** Representative images of microglia (Iba-1 positive, red) engulfing pre-synaptic terminals immunolabeled for pre-synaptic marker synaptophysin (SYP, green) in the hippocampus of mock- (**a**) or ZIKV-infected mice (**b**) at 6 dpi. White arrows point to SYP-positive elements engulfed within microglia. Scale bar = 20 µm. **c** Quantification of microglia-SYP colocalization (t = 3.502, *p = 0.0128; N = 3 (mock) and 5 (ZIKV)). **d–n** Mice were treated with minocycline (Mino; 50 mg/kg i.p., 3 times/week beginning one week prior to infection). **d–f** Representative images of Iba-1 immunoreactivity in hippocampi of mock- (**d**), ZIKV-infected (**e**), or minocycline-treated ZIKV-infected mice (**f**) mice at 10 dpi. Scale bar = 20 µm. **g** Iba-1 positive cells in mice hippocampi at 10 dpi. **h** Maximum-minimum Feret index from microglia in hippocampi of mice (**g**: F(2,10) = 3.986; *p = 0.005 mock vs ZIKV, p = 0.0145 mock vs ZIKV+Mino; **h**: F(2,10) = 3.960; *p < 0.0001 mock vs ZIKV and ZIKV. vs. ZIKV+Mino, one-way ANOVA followed by Bonferroni). (N = 4 (mock), 4 (ZIKV), 5 (ZIKV+Mino)). **i** Mice were tested in the novel object recognition (NOR) test at 9 dpi (i: t = 2.736 *p = 0.0144 for mock; t = 2.578 *p = 0.0366 for ZIKV+Mino, one-sample Student's t-test compared to 50% chance level; N = 8 (Mock), 8(ZIKV+Mino), and 10 (ZIKV); data represent one of two independent experiments). **j–n** Minocycline prevents ZIKV-induced impaired LTP in dorsal hippocampal CA3 area. **j** Means ± SEM of slopes of field excitatory post-synaptic potentials (fEPSPs) evoked in CA3 by mossy fiber stimulation, before and after high-frequency stimulation (HFS), in slices from mice treated with minocycline. **k** Representative fEPSPs recorded before (black) and 35–40 min after (red) HFS. **l** Average fEPSP slopes 35–40 min post-HFS. **m** Baseline paired-pulse amplitude ratios. (N = 8 slices from 5 mock-infused and 10 slices from 7 ZIKV-infected mice). **n** TNF-α expression in hippocampi of mice at 10 dpi (F(2,12) = 2.538, *p = 0.0005 mock vs. ZIKV+Veh or ZIKV+Mino, one-way ANOVA followed by Bonferroni's, N = 5 mice/group). Bars represent means ± SEM. Symbols represent individual mice (**a–l**, **n**) or individual slices (**l–m**). Source data from panels **c**, **g–n** are provided as Source Data File

mature neural cells and adult human brain tissue are susceptible to infection by ZIKV remained to be determined.

Addressing this gap, we report that ZIKV replicates in ex vivo slices from adult human cortical tissue. We detected infectious viral particles released to the medium, indicating successful assembly of particles capable of spreading infection. Using immunocompetent mice infused via i.c.v. with ZIKV, we further showed that ZIKV replicates in the adult rodent brain, reaching maximal viral titers 6 days after inoculation. A recent study showed viral replication in the brain in a similar time frame following systemic administration of an Asian ZIKV strain to immunocompromised mice[25]. Intriguingly, in that study viral replication was mainly detected at sites of adult neurogenesis. In contrast, we show replication of the Brazilian ZIKV strain in slices from human adult temporal lobe, a region not considered a classical neurogenic niche. We further show that, in adult mice, the highest viral loads were detected in the hippocampus and

frontal cortex, brain regions closely related to cognition, memory and executive function, and in the striatum, a structure involved in motor control. These findings strengthen the usefulness of our model to unravel the consequences of ZIKV infection in the mature brain.

In contrast with the brain regional selectivity for viral replication verified following i.c.v. infection in wild-type mice, we found that high and similar viral titers were detected throughout the brain when mice deficient in type I interferon receptors were infected with ZIKV. Similar results were found following systemic infection by ZIKV in wild-type mice at post-natal day 3, a stage when interferon-mediated inflammatory response is inefficient. It appears likely, therefore, that lack of the ability to mount an effective interferon-based response leads to generalized replication of ZIKV across the brain, abolishing the preferential targeting of regions associated with cognitive and motor control verified in immunocompetent adult mice. This suggests that the

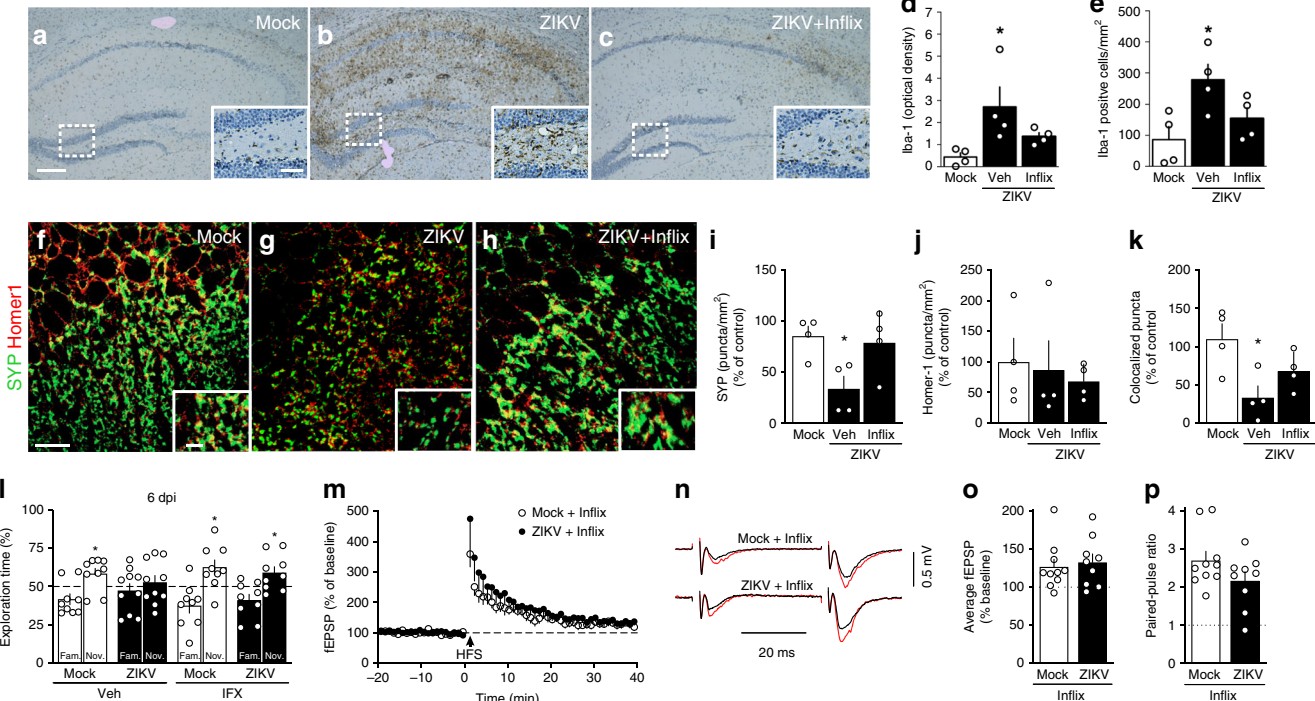

**Fig. 6** TNF-α mediates microglial activation, ZIKV-induced synapse and memory impairment. Adult Swiss mice received an i.c.v. infusion of $10^5$ PFU ZIKV or mock medium and were treated (or not) with Infliximab (Inflix; 0.2 μg/day, from 2 to 6 dpi, i.c.v.). **a–c** Representative images of Iba-1 immunolabeling in hippocampi of mock-infused (**a**), ZIKV-infected mice treated with vehicle (**b**), or ZIKV-infected mice treated with Infliximab (**c**) at 10 dpi. Scale bar = 200 μm, inset scale bar = 50 μm. **d, e** Integrated Iba-1 optical density (**d**) and number of Iba-1-positive cells (**e**) in the hippocampus (**d** $F_{(2,9)}$ = 4.629, *p = 0.043; **e** $F_{(2,9)}$ = 5.254, *p < 0.0327, one-way ANOVA followed by Bonferroni, N = 4 mice/group). **f–h** Representative images of the CA3 hippocampal region of mock-infused (**f**), ZIKV-infected mice treated with vehicle (**g**), or ZIKV-infected mice treated with Infliximab (**h**) immunolabeled for synaptophysin (SYP, green) and Homer1 (red). Scale bar = 25 μm, inset scale bar = 5 μm. **i–k** SYP (**i**), Homer1 (**j**), and colocalized SYP/Homer1 puncta (**k**). (**i** $F_{(2,9)}$ = 4.917, *p = 0.0364, **k** $F_{(2,9)}$ = 5.682, *p = 0.0166, one-way ANOVA followed by Bonferroni's; N = 4 mice/group). **l** Mice were tested in the NOR test at 6 dpi (**l**: t = 2.364 *p = 0.0456 for mock/Veh, t = 2.688 *p = 0.0276 for mock/Inflix, t = 2.310 *p = 0.0497 for ZIKV/Inflix; one-sample Student's t-test compared to the chance level of 50%; N = 10 (ZIKV) and 9 (other groups); data represent one of two independent experiments). **m–p** Infliximab prevents ZIKV-induced impairment in LTP in dorsal hippocampal CA3 area. **m** Means ± SEM of slopes of field excitatory post-synaptic potentials (fEPSPs) evoked in CA3 by mossy fiber stimulation, before and after high-frequency stimulation (HFS), in slices from mock- or ZIKV-infused mice treated with infliximab. **n** Representative fEPSPs before (black) and 35–40 min after (red) HFS. **o** Average fEPSP slopes 35–40 min post-HFS. **p** Baseline paired-pulse amplitude ratios. (N = 10 slices from 7 mock-infused and 9 slices from 6 ZIKV-infected mice). Bars represent means ± SEM. Symbols correspond to individual mice (**a–l**) or individual slices (**o–p**). Source data (panels **d–e**, **i–p**) are provided as Source Data File

brain region-specificity of the host immune response may be an important determinant for the efficiency of viral replication in different brain structures.

We additionally note that we detected ZIKV RNA (albeit at lower levels than following i.c.v. infection) in frontal cortex and hippocampus after intravenous inoculation of ZIKV in wild-type adult mice. These findings demonstrate that ZIKV targets memory-related brain regions regardless of whether the route of infection is central or peripheral. They further underline the translational potential of the adult ZIKV infection model used here, since the virus has been detected in CSF of adult patients with neurological complications[10–12,18,22].

An early study by Bell and colleagues (1971) concluded that both neurons and astrocytes were infected by the African strain of ZIKV adapted to mice, leading to hippocampal degeneration and necrosis 7 days post-infection[38]. Moreover, a previous report showed that NPCs were the main cell type targeted by ZIKV in adult immunocompromised mice[25]. We found no NS2B immunostaining in doublecortin-positive cells at the peak of replication. However, we found a reduced number of doublecortin-positive cells in the hippocampal dentate gyrus of ZIKV-infected mice. It is, thus, possible that ZIKV targets NPCs at early stages of infection, leading to cell death and to lack of detection of NS2B immunostaining in NPCs.

Our results indicate that neurons are the main cell type infected by ZIKV in the adult mouse and human brain. Current findings are in contrast with those reported in the developing brain[3], when ZIKV infects both neurons and glia, suggesting that ZIKV may target distinct cell populations at different stages of brain development. Although our results demonstrate that ZIKV infects neurons, how these cells react to viral infection remains to be elucidated. Previous studies showed that ZIKV-infected neurons activate intracellular innate immune pathways, leading to the release of proinflammatory cytokines[39,40]. Although the mechanisms linking neuronal infection and cytokine production are still a matter of debate, previous studies along with our current results suggest that neurons are both a target of ZIKV infection and a potential source of inflammation.

Viral infections have been associated with neurological damage in humans and mice. Learning and memory are frequently affected, with symptoms often persisting for long periods after the acute phase of the disease[31,41]. We found that brain infection by ZIKV caused memory impairment in adult mice, and that this was accompanied by reduced synaptic density in hippocampal

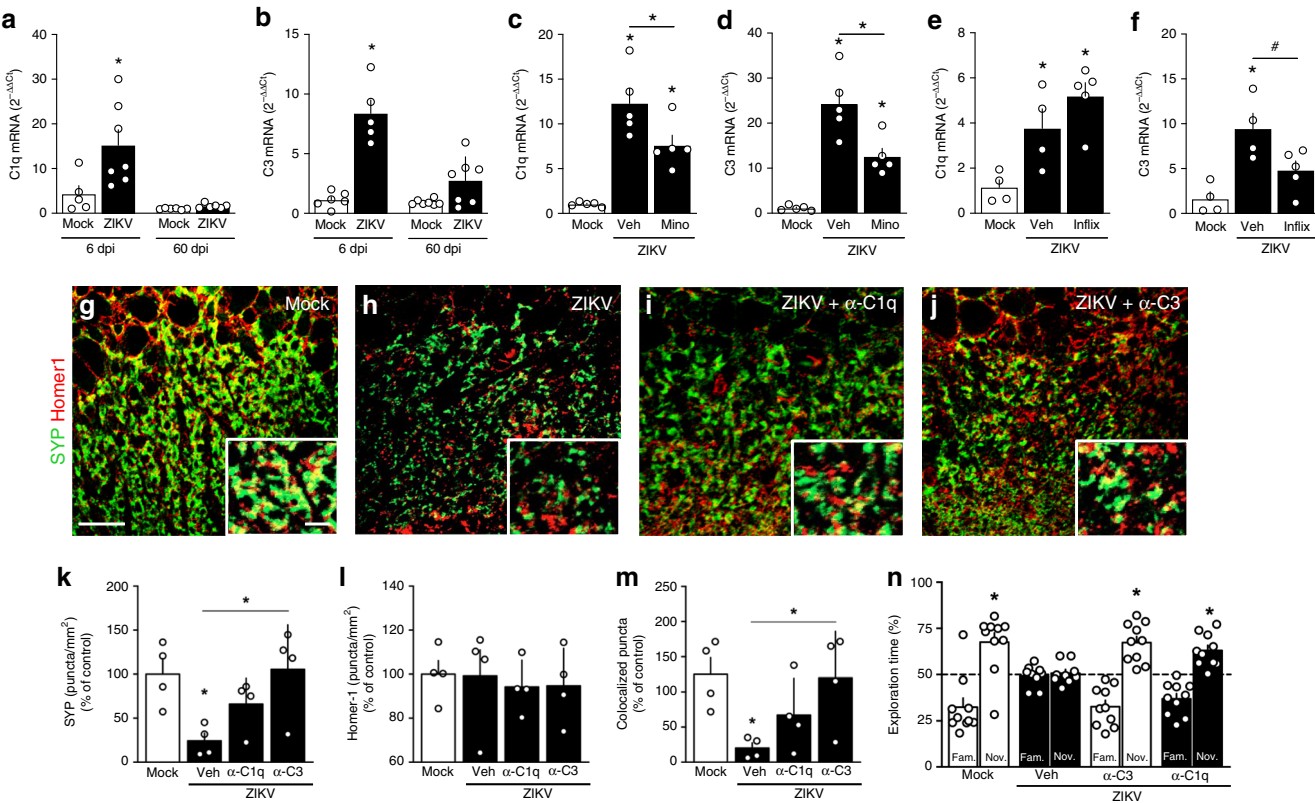

**Fig. 7** Role of complement system in ZIKV-induced memory impairment. **a**, **b** Adult mice received an $10^5$ PFU ZIKV (i.c.v.) or mock medium. At 6 or 60 dpi, brain expression of C1q (**a** $F_{(3,22)} = 11.63$, $*p = 0.003$; one-way ANOVA followed by Bonferroni's) and C3 (**b** $F_{(3,21)} = 26.92$, $*p < 0.0001$; one-way ANOVA followed by Bonferroni's) were measured. $N = 6$ (6 dpi, mock), 5 (6 dpi, ZIKV), and 7 (60 dpi, both groups). **c–f** Mock-infused and ZIKV-infected mice were treated with minocycline (**c**, **d** Mino; 50 mg/kg i.p., 3 times/week beginning 1 week prior to infection) or Infliximab (**e**, **f** Inflix; 0.2 μg/day, from 2 to 6 dpi, i.c.v.). At 6 dpi, brain expression of C1q (**c** $F_{(2,12)} = 22.84$; $*p < 0.0001$ mock vs. ZIKV; $*p = 0.0054$ mock vs. ZIKV+Mino; $*p = 0.0388$ ZIKV vs. ZIKV+Mino; $N = 5$ mice/group. **e** $F_{(1,10)} = 10.49$; $*p = 0.038$ ZIKV; $*p = 0.0021$ ZIKV+Inflix, one-way ANOVA followed by Bonferroni, $N = 4$ (mock and ZIKV+veh) and 5 (ZIKV+Inflix)) and C3 (**d**: $F_{(2,12)} = 29.63$, $*p < 0.0001$ mock vs. ZIKV, $*p = 0.0075$ mock vs. ZIKV+Mino; $*p = 0.0064$ ZIKV vs. ZIKV +Mino; $N = 5$ mice/group. f: $F_{(2,10)} = 9.464$, $*p = 0.0045$ mock vs. ZIKV, $\#p = 0.0674$ ZIKV vs. ZIKV+Inflix; one-way ANOVA followed by Bonferroni, $N = 4$ (mock and ZIKV+veh) and 5 (ZIKV+Inflix). **g–m** Mock-infused and ZIKV-infected mice treated with PBS or 0.15 μg antibodies against C3 or C1q at 4 or 5 dpi. **g–j** Representative images of the CA3 hippocampal region of mock-infused (**g**), ZIKV+veh (**h**), ZIKV+anti-C1q antibody (**i**), and ZIKV+anti-C3 antibody mice (**j**) immunolabeled for synaptophysin (SYP, green) and Homer1 (red). Scale bar = 25 μm, inset scale bar = 5 μm. **k–m** Puncta for SYP (**k**), Homer1 (**l**) and colocalized SYP/PSD-95 (**m**). (**k** $F_{(3,12)} = 4.514$, $*p = 0.031$ mock vs. ZIKV, $p = 0.0205$ ZIKV vs. ZIKV/α-C3. **m** $F_{(3,12)} = 4.077$, $*p = 0.0317$ mock vs. ZIKV, $p = 0.0412$ ZIKV vs. ZIKV/α-C3, one-way ANOVA followed by Bonferroni, $N = 4$ mice/group). **n** Mock-infused and ZIKV-infected mice treated with anti-C3, anti-C1q (0.30 μg, i.c.v.) or vehicle (PBS). At 6 dpi, mice were tested in the NOR test (t = 3.527 $*p = 0.0064$ Mock+veh, t = 5.120 $*p = 0.0006$ ZIKV+α-C3; t = 4.679 $*p = 0.0012$ ZIKV+α-C1q; one-sample Student's $t$-test compared to 50% chance level, $N = 9$ (ZIKV+veh), 10 (other groups). Bars represent means ± SEM. Symbols represent individual mice. Source data from panel **a–f** and **k–n** are provided as Source Data File

CA3 and DG. Our results further indicate that memory impairment induced by ZIKV was not caused by sickness behavior. Results also showed that not all arboviruses cause memory impairment, as mice infected by MAYV had no memory dysfunction. Recent studies have demonstrated memory impairment and synapse pathology in mice following infection by other neuroinvasive viruses, such as HIV[42], WNV[31], Herpes Simplex virus-1[43], and Borna disease virus[44].

We further demonstrate that phosphorylated CREB, an important transcription factor involved in memory mechanisms[30], was markedly decreased in the hippocampus and cortex of ZIKV-infected mice. Along with synapse damage, impaired CREB phosphorylation/signaling is centrally implicated in synapse and memory dysfunction[30]. Electrophysiological recordings of fEPSPs evoked through the mossy fiber pathway in hippocampal slices from ZIKV-infected mice showed that both PTP and LTP at DG-CA3 synapses in the dorsal hippocampus were significantly

inhibited. Moreover, and consistent with a pre-synaptic impact of ZIKV at CA3, paired-pulse facilitation at DG-CA3 synapses was inhibited in slices from ZIKV-infected mice.

Reactive gliosis has been described in developing brains of mice[32] and in humans exposed to ZIKV in uterus or during adulthood[3,18,22]. We demonstrate that ZIKV infection triggered marked neuroinflammation and brain microgliosis in adult mice. Microglia play a central role in the pruning of synapses tagged by complement system proteins in a mouse model of WNV infection[31]. Significantly, we found that treatments with infliximab or minocycline prevented memory and synapse dysfunction in ZIKV-infected mice. Brains of ZIKV-infected mice showed increased expression of complement system proteins, C1q and C3. Further implicating aberrant activation of TNF-α, microglia and complement in ZIKV-instigated synapse and memory deficits, blockade of TNF-α signaling prevented microglial activation and synapse damage, as well as upregulation of C3 in ZIKV-

infected mice. We note that complement in the CNS in ZIKV-infected mice may originate from both disruption of BBB integrity and infiltration of peripheral immune cells and from activation of resident microglia. These possibilities are supported by our finding that both central and peripheral treatments with infliximab resulted in decreased brain expression of C3. However, the lack of effect of infliximab on brain C1q expression in ZIKV-infected mice may be related to the capacity of neurons to express C1q[45,46]. Importantly, we show that i.c.v. infusion of either anti-complement antibodies preserved synapses and memory in ZIKV-infected mice, indicating a central role of complement activation in ZIKV-induced dysfunction.

In conclusion, we have established that ZIKV infects and replicates in adult human brain tissue. Further, our findings identify a possible mechanism underlying neurological complications in adults infected with ZIKV[8–23]. Results revealed that ZIKV caused brain inflammation and that elevated TNF-α led to microglial activation and phagocytosis of hippocampal synapses in mice. This culminates in synapse dysfunction and memory impairment, in the absence of overt neurodegeneration. This is consistent with the observation that memory impairment was reversible upon resolution of the infection and inflammation, suggesting that cognitive deficits are caused by aberrant signaling mechanisms triggered by infection rather than by neuronal death.

Although the initial wave of the recent ZIKV epidemics in the Americas is now past, the fact that mosquito populations are not under control suggests that the number of infected individuals will likely remain elevated in coming years, and may eventually spread to additional geographical areas. From a public health perspective, it would be important to assess memory and cognition in infected adults as a potential significant comorbidity of ZIKV infection. Future studies aimed to examine the long-term neurodegenerative impact of ZIKV infection are warranted, not only in developing brains but also in the adult CNS. Identifying ways to prevent signaling abnormalities leading to brain dysfunction could pave the way to intervention strategies to delay or prevent the development of major neurological conditions in ZIKV-infected individuals.

## Methods

**Viruses**. ZIKV, isolated from a febrile case in the State of Pernambuco, Brazil, was kindly provided by Dr. Ernesto T. A. Marques Jr. (Centro de Pesquisas Aggeu Magalhães, Fiocruz, Pernambuco; gene bank ref. number KX197192). Samples were prepared as described elsewhere[47]. Briefly, C6/36 mosquito cells were infected with a multiplicity of infection (MOI) of 0.01 for 7 days. Following infection, the culture medium was centrifuged at $700 \times g$ for 10 min, and the supernatant was stored at −80 °C. For determination of viral titers, plaque assays were performed in Vero cells treated with serial dilutions of stocks or samples. Infected Vero cells were maintained in Dulbecco's Modified Eagle's Medium (DMEM) containing 1% FBS, 1 U/mL penicillin/streptomycin (LGC Biotecnologia; São Paulo, Brazil), and 1.5% carboxymethylcellulose (CMC; Sigma) and after 5 days cells were fixed with 4% formaldehyde for 15 min, stained with crystal violet (1% crystal violet in 20% methanol), and the number of plaques was counted. Medium harvested from non-infected C6/36 cells were used as control (mock). In some experiments, an additional control group consisting of ZIKV inactivated by exposure to UV-light (30 W) for 40 min was used (distance from UV-source: 20 to 25 cm; sample volume: 300–500 μl). Successful inactivation was confirmed by plaque assay. Mayaro virus (ATCC VR 66, strain TR 4675) was propagated in BHK-21 (ATCC-CCL-10) with a multiplicity of infection (MOI) of 0.1 for 24 h. Viral titer of stocks and samples was determined by plaque assay using Vero cells.

**Adult human cortical slices**. Adult human cortical slices were prepared and characterized as described[48], with minor modifications. All procedures were approved by the National Committee for Research Ethics of the Brazilian Ministry of Health (protocol #0069.0.197.000-05). Samples of temporal lobe cortical tissue were obtained from patients with drug-resistant epilepsy subjected to temporal lobectomy. Informed consent was obtained from all participants. Surgeries followed the international standard for anterior lobectomy, consisting of removal of tissue 3–4 cm from the pole of the temporal lobe, and concomitant removal of the hippocampus (where epileptic foci were located). Samples used in the current study comprised temporal lobe cortical fragments plus associated subcortical tissue. After

dissection under sterile conditions, 400 μm-thick slices were prepared using a McIlwain tissue chopper and plated on Neurobasal A medium containing 2% B27 supplement, 500 μM glutamine, 5 ng/mL FGF2, 2 μM DHEAS, 1 ng/mL BDNF, supplemented with 50 μg/ml gentamicin (Sigma). After 4 days in culture at 37 °C and 5% $CO_2$, slices were exposed to $10^7$ PFU ZIKV (or an equivalent volume of mock medium) in a total volume of 200 μl. After 1 h, virus was washed out with 500 μL of warm Neurobasal A medium, and maintained in fresh medium at 37 °C/5% $CO_2$ for 4, 24, 48, or 72 h until use in subsequent analyses. Medium was harvested at specified time points for determination of viral titers by plaque assay. For immunohistochemical detection of ZIKV, tissue sections were fixed in 4% paraformaldehyde 24 h after exposure to ZIKV or mock medium, as described above.

**Viral infections**. Male Swiss mice, male C57BL/6 mice or male $Il1r^{-/-}$ mice from our animal facility were 2.5–3-month-old at the beginning of experiments. A129 mice were 5–8-weeks-old at the beginning of experiments. Animals were housed in groups of five per cage with free access to food and water, under a 12 h light/dark cycle, with controlled temperature and humidity. All procedures followed the "Principles of Laboratory Animal Care" (US National Institutes of Health) and were approved by the Institutional Animal Care and Use Committee of the Federal University of Rio de Janeiro (protocols #043/2016 and #126/2018). For infusion into the lateral ventricle (i.c.v.), animals were anesthetized with 2.5% isoflurane (Cristália; São Paulo, Brazil) using a vaporizer system (Norwell, MA) and were gently restrained only during the injection procedure. As previously described, a 2.5 mm-long needle was unilaterally inserted 1 mm to the right of the midline point equidistant from each eye and parallel to a line drawn through the anterior base of the eye[49]. Three microliter of ZIKV ($10^5$ or $10^3$ PFU), UV-inactivated ZIKV or mock medium (as indicated in Figure Legends) were slowly infused using a Hamilton syringe. In some experiments, mice received an i.c.v. infusion of 3 μL of MAYV ($10^5$ PFU). Mice showing any signs of misplaced injections or brain hemorrhage (~5% of animals throughout our study) were excluded from further analysis. Infusion into the third brain ventricle was performed in an independent group of animals (see "Results"). Before procedure, adult male Swiss mice were anesthetized (90 mg/kg ketamine and 4.5 mg/kg xylazine, i.p.) and had their heads secured in a stereotaxic frame (Kopf Instruments). The skin and periosteum were removed to expose the skull, and ZIKV ($10^5$ PFU) or mock medium were stereotaxically injected into the third ventricle (3 μL; 0.18 mm anteroposterior, 0 mediolateral, 0.5 mm dorsoventral from Bregma) using a Hamilton syringe. After surgery, the skull was sealed with dental cement, and animals were euthanized 6 days post-infection (dpi).

In one experiment, neonatal (post-natal day 3) Swiss-mice were submitted to subcutaneous (s.c.) injection of $10^6$ PFU of ZIKV. Pups were then killed by decapitation at 12 dpi, the peak of viral replication at this age[32].

**Brain replication of MAYV in mice**. Brains were collected at different time points following infection, homogenized to a concentration of 0.5 mg tissue/μL in DMEM, and serially diluted for determine of viral load by plaque assay using Vero cells. Viral load was expressed as PFU/mg brain tissue (PFU/mg).

**Pharmacological treatments**. For TNF-α blockade, infliximab (Remicade; Janssen-Cilag/Switzerland) was injected either i.p. or i.c.v., as indicated. For systemic treatment, mice received daily injections of 20 μg infliximab in 200 μl PBS (137 mM NaCl, 10 mM sodium phosphate, 2.7 mM KCl, pH 7.4) or an equal volume of PBS beginning on the day of infection. For i.c.v. treatment, mice received infusions of 0.2 μg infliximab (in 3 μL PBS) or an equal volume of PBS during 4 consecutive days beginning 48 h after ZIKV infection. For blockage of microglia activation, mice received injections of minocycline hydrochloride (50 mg/kg, Sigma–Aldrich) or an equal volume of saline via intraperitoneal route (i.p.), three times/week during 2 weeks. This treatment began the week before ZIKV infection and was discontinued 48 h before behavioral analysis. For blockage of brain complement proteins, mice received i.c.v. injections of 0.15 μg (in 3 μL) of antibodies against C3 (Abcam, #11862), C1q (Abcam, #11861), or the equivalent volume of PBS on days 4 and 5 post-infection.

**Tissue collection and preparation**. Mice were anesthetized (90 mg/kg ketamine and 4.5 mg/kg xylazine, i.p.) before euthanasia. Brains (without cerebellum), isolated brain structures and/or peripheral organs were collected, immediately frozen in liquid nitrogen and stored at −80 °C before protein or RNA extraction.

**RNA extraction and qPCR**. Tissues were homogenized to a concentration of 0.5 mg tissue/μL in DMEM, and 200 mL of the homogenate were used for RNA extraction with Trizol (Invitrogen), according to manufacturer's instructions. RNA purity and integrity were determined by the 260/280 and 260/230 nm absorbance ratios and by agarose gel electrophoresis. Only preparations with absorbance ratios >1.8 and no signs of RNA degradation were used. One microgram of total RNA was used for cDNA synthesis using the High-Capacity cDNA Reverse Transcription Kit (ThermoFisher Scientific Inc). Analyses were carried out on an Applied Biosystems 7500 RT–PCR system using the TaqMan Mix (ThermoFisher Scientific) kit according to manufacturer's instructions. Primers for ZIKV were used as

described[50]: forward, 5′-CCGCTGCCCAACACAAG-30; reverse, 5′-CCAC-TAACGTTCTTTTGCAGACAT-3′; probe, 5′-/56-FAM/AGCCTACCT/ZEN/TGACAA GCAATCAGACACTCAA/3IABkFQ/-3′ (Integrated DNA Technologies). For negative strand quantification, cDNA was synthesized using 2 pM of ZIKV 835 forward primer, 5′-TTGGTCATGATACTGCTGATTGC-3′[51] instead of random primers. Cycle threshold (Ct) values were used to calculate the equivalence of $\log_{10}$ PFU/mg tissue after conversion using a standard-curve with serial 10-fold dilutions of ZIKV stock.

For cytokine and complement system protein expression, qPCR analyses were performed using the Power SYBR kit (Applied Biosystems; Foster City, CA). Actin was used as an endogenous control. Primer sequences were the following: IL-1β Forward: 5prime;-GTA ATG AAA GAC GGC ACA CC-3′ and Reverse: 5′-ATT AGA AAC AGT CCA GCC CA-3′; TNF-α Forward: 5′-CCC TCA CAC TCA GAT CAT CTT CT-3′ and Reverse: 5′-GCT ACG ACG TGG GCT ACA G-3′; Actin Forward: 5′-TGT GAC GTT GAC ATC CGT AAA-3′ and Reverse: 5′-GTA CTT GCG CTC AGG AGG AG-3′; C1q Forward: 5′-CTC AGG GAT GGC TGG TGG CC-3′ and Reverse: 5′-CCT TTG AGA CCC GGC CTC CC-3′; C3 Forward: 5′-ACC CAG CTC TGT GGG AAG TG-3′ and Reverse: 5′-CTT CAT AGA CTG CTG CAA CCA-3′.

**Immunohistochemistry**. Mice were transcardially perfused with cold PBS followed by ice-cold 4% paraformaldehyde. Brains were removed and post-fixed for 24 h in the same solution, and embedded in paraffin after dehydration and diaphanization. For immunohistochemistry, paraffin-embedded brain tissue sections (3–5 μm) were immersed in xylene for 10 min, rehydrated in absolute ethanol followed by 95% and 70% solutions of ethanol in water, and incubated with 3% $H_2O_2$ in methanol for inactivation of endogenous peroxidase. Antigens were reactivated by treatment with 0.01 M citrate buffer for 50 min at 95 °C. Slides were washed in PBS and incubated with primary antibodies (rabbit anti-GFAP, 1:500, DAKO Z0334; rabbit anti-Iba-1, 1:1,000, Wako NCNP24; rabbit anti- CREB pSer[133], 1:100, Cell Signaling #9198; rabbit anti-NS2B, 1:50, GeneTex #133308) diluted in immuno-histochemistry antibody diluent (Enzo Life Siences) overnight at 4 °C. After washing with PBS, slides were incubated with biotinylated secondary antibodies for 1 h at room temperature, washed twice with PBS and incubated with streptavidin-biotin-peroxidase (Vector Laboratories, CA) for 30 min. Slides were then covered with 3,3′-diaminobenzidine solution (0.06% DAB in PBS containing 2% DMSO and 0.018% $H_2O_2$) for 1 min. Identical conditions and reaction times were used for slides from different animals (run in parallel) to allow comparison between immunoreactivity optical densities. Reactions were stopped by immersion of slides in distilled water. Counter-staining was performed with Harris hematoxylin. Slides were imaged using a Sight DS-5M-L1 digital camera (Nikon, Melville, NY) connected to an Eclipse 50i light microscope (Nikon). For GFAP, Iba-1, and phospho-CREB quantification, one image from each hippocampal subfield (CA1, hilus and DG) in each hemisphere and two images from the parietal cortex in each hemisphere were obtained using ×200 magnification, and an optical density threshold that best discriminated staining from background was defined using NIH ImageJ as described[52]. Total pixel intensity was determined for each image and data are expressed as integrated optical density (OD). Alternatively, the number of Iba-1-positive cells per $mm^2$ was counted. For morphological characterization of Iba-1-positive cells, we analyzed cell bodies in two dimensions and measured Feret diameters as previously described[53]. Maximum and minimum Feret are defined as the longest and shotest diameters of a cell body, respectively. In a spherical cell body, max-min Feret differences tend to zero.

For immunofluorescence, mouse brains or human tissue slices (40 μm thick) were fixed in 4% PFA for 24 h, maintained in 30% sucrose for 48 h and embedded in optimal cutting temperature compound (OCT) for cryostat sectioning (Leica). Slides with mouse coronal brain sections or human tissue were fixed in acetone for 30 min, washed twice with PBS and incubated with primary antibodies (rabbit anti-NS2B, 1:50, GeneTex #133308; mouse anti-GFAP, 1:500, Sigma #G3893; rat anti-F4/80, 1:25, Bio-Rad #MCA497; mouse anti-NeuN, 1:50, Chemicon #MAB377; rabbit monoclonal anti-TMEM119, 1:50, Abcam #210405; mouse anti-DCX 1:50, Santa Cruz #271390, mouse anti-synaptophysin 1:200, Vector Laboratories #S285; rabbit anti-Homer-1 1:100, Abcam #184955) diluted in PBS containing 3% BSA. Sections were then incubated with Alexa 555-, 594- or 488-conjugated secondary antibodies (1:750; Invitrogen) for 1 h at room temperature, washed in PBS and mounted in Prolong Gold Antifade with DAPI (Invitrogen). Slides were imaged on an Axio Observer Z1 microscope (Zeiss) equipped with an Apotome module or on a confocal microscope (Nikon) at ×630 magnification. Synaptic and microglia (Iba-1) puncta were imaged on a confocal microscope (Nikon) at ×630 magnification. Three independent images of each brain structure were used for analyses. Each image acquired was a z-stack of 12–16 (0.33 μm depth) sections. For synaptic puncta quantification, maximum projections of three consecutive optical sections (corresponding to 1 μm total depth) were generated from the middle of the original z-stack. We used the Puncta Analyzer plugin for ImageJ 1.29 (NIH; RRID: SCR_003070) to count the number of colocalized, pre- (synaptophysin), or post-synaptic (Homer-1) puncta. Averaged puncta values for mock-infused mice (controls) were used to calculate percentage of puncta results for all groups[54].

To determine synapse engulfment by microglia, three confocal Z-stack images from the CA3 hippocampal region from ZIKV-infected or mock-injected mice

were acquired using a Nikon CII confocal microscope. Each image comprised 12–18 0.3 μm optical planes, three of which were analyzed independently as previously described[52]. Briefly, red (IBA-1) and green (synaptophysin) channels were processed separately on FIJI. Background from the red channel was subtracted using a 50 pixel radius rolling ball and then subjected to a 2 pixel radius 3D median filter in every dimension. Background and noise from the green channel were subtracted using a 2 pixel rolling ball radius followed by the Despeckle function. Images were then filtered through a 3D maximum filter (radius 3 pixels) in every dimension, auto-thresholded using the FIJI default method and processed using the "watershed" function. The two channels were then merged for puncta analyses using Puncta Analyzer (v2.0 plugin, NIH Image J), using a 50 pixel rolling ball, threshold levels 40 in the red channel and 65 in green channel. Only synaptophysin puncta bigger than 10 pixels[2] in size and contained within Iba-1-positive cells were quantified. The number of colocalized synaptophysin (Syp) and Iba-1 puncta was normalized by the number of Iba-1 puncta in each plane. Data are expressed relative to mock-infused mice. Representative images show three-dimensional reconstructions (with corresponding orthogonal views, xz and yz) obtained from three optical planes from the original raw image. 3D-Videos from confocal stack images were made in FIJI.

**FluoroJade B (FJ) staining**. FJ histochemistry was used as indicative of neuronal degeneration. Paraffin-embedded brain tissue sections were sequentially immersed in 100% ethanol for 3 min, 70% ethanol for 1 min, and distilled water for 1 min. Sections were then immersed in 0.06% potassium permanganate for 10 min (to suppress endogenous background signal), and washed with distilled water for 1 min. FJ B staining solution (10 mL of 0.01% FJ aqueous solution added to 90 mL of 0.1% acetic acid in distilled water) was added for 30 min. After staining, sections were rinsed three times in distilled water. Excess water was drained off, and slides were coverslipped with dibutylphthalate in xylene (D.P.X.) mounting medium (Aldrich Chem. Co.). Sections comprising the hippocampus and parietal cortex were imaged on epifluorescence microscopes (Olympus—Bx41 or Nikon Eclipse 50i)[50]. Positive neurodegeneration staining controls consisted of sections from the hippocampus of a mouse injected i.c.v. with 36.8 nmol quinolinic acid and euthanized 24 h thereafter.

**Behavioral tests**. *Novel object recognition test*: The test was carried out in an arena measuring 30 × 30 × 45 cm. Test objects were made of plastic and had different shapes, colors, sizes, and textures. During behavioral sessions, objects were fixed to the box using tape to prevent displacement caused by exploratory activity of the animals. Preliminary tests showed that none of the objects used evoked innate preference. Before training, each animal was submitted to a 5 min-long habituation session, in which it was allowed to freely explore the empty arena. Training consisted in a 5 min-long session during which animals were placed at the center of the arena in the presence of two identical objects. A trained researcher recorded the amount of time mice spent exploring each object. Sniffing and touching the object were considered exploratory behavior. The arena and objects were cleaned thoroughly with 70% ethanol in between trials to eliminate olfactory cues. Two hour after training, animals were again placed in the arena for the test session, with one of the objects used in the training session having been replaced by a new one. Again, the amount of time spent exploring familiar and novel objects were measured. Results were expressed as percentage of time exploring each object during the training or test sessions, or as total exploration during each session. Data were analyzed using a one-sample Student's *t*-test comparing the mean exploration percentage time for each object with the chance value of 50%. Animals that recognize the familiar object as such (i.e., learn the task) explore the novel object >50% of the total time[55].

*Open field task*: For the open field test, animals were placed at the center of an arena (30 × 30 × 45 cm) divided into nine equal quadrants by imaginary lines on the floor. Total locomotor activity, number of rearings, number of body rotations, and time spent at the center or at the periphery of the arena were recorded for 30 min using ANY-maze software (Stoelting Company). In some experiments (Supplementary Fig. 3a, f) shorter (5 min) open field sessions were performed, and both distance traveled and time at the center of the arena were recorded. The arena was thoroughly cleaned with 70% ethanol in between trials to eliminate olfactory cues. The amount of time spent by an animal in the central area of the arena is considered to be inversely proportional to anxiety[56].

*Elevated plus maze task*: The plus maze apparatus consisted of a platform comprising two open and two closed arms (each 4.5 cm wide, 30 cm in length), connected by a central square (6 × 6 cm), elevated 60 cm from the ground. Mice were individually placed at the center of the apparatus facing one of the closed arms. During the 5-min-long sessions, the amount of time each animal spent in the open or closed arms of the maze was determined using ANY-maze software[56].

*Passive avoidance (step-down) test*: The step-down apparatus consisted of a gridded floor box (40 × 25 × 35 cm) with a platform (9.5 × 7 × 3 cm) at the center. During the training session, animals were placed in the platform and time for stepping down was recorded. Once the animals placed the four paws on the metallic grid, an electric shock of 0.8 mA was applied for 2 s. The test session took place 24 h after training, when animals were again placed on the platform and latency to step-down from it was measured. Data are expressed as latencies to step-down from platform during training and test sessions, or as retention score (step-

down latency in the training session subtracted from latency in the test session for each animal).

*Rotarod*: This was performed in the rotarod apparatus for mice (EFF 412; Insight Ltda., Brazil), using a protocol adapted from Fortuna et al. (2017), which consisted in one habituation session and one test session. Briefly, mice were individually placed in the apparatus floor for 3 min and then on the aluminum cylinder without rotation for another 2 min for habituation. The test phase consisted of three identical trials (intertrial interval time: 60 min), in which animals were placed on top of the rod rotating at increasing speed (minimum speed: 16 rpm; maximum speed: 36 rpm; acceleration: 3.7 rpm$^2$). Latency to fall from the rotating rod was determined, and if mice did not fall from the rod in 5 min, they were removed and placed back into their home cages. Results are shown as averages of latencies to fall in test trials.

*Mechanical sensitivity*: Mechanical nociceptive thresholds in mice were measured using von Frey filaments (Stoelting, Chicago, IL)[57]. Briefly, six filaments with increasing diameters were applied 10 times each, with 2 s interval in between, to slightly varying loci in the plantar surface of both hind paws. Removal of the paw in response to the filament was considered a positive response, and frequencies of positive responses were calculated. Mechanical baseline measures were obtained from naïve animals one week before ZIKV infusion.

*Thermal sensitivity*: Thermal heat sensitivity was evaluated by a modified Hargreaves method[57]. Animals were placed in a plexiglas apparatus ($7 \times 9 \times 11$ cm) on an elevated surface and allowed to habituate for 90 min before testing. A heat source was directed to the plantar surface of each hind paw and the latency to withdraw the paw was measured. Infrared intensity was adjusted to obtain basal paw-withdrawal latencies of 7–10 s (as established by measurements before ZIKV infusion). An automatic 20 s cutoff was used to prevent tissue damage. Cold sensitivity was measured by the acetone drop method. For these tests, animals were placed in a plexiglass box and were allowed to freely explore it for 90 min. Then, mice received a drop of 30% acetone solution in one paw. Nocifensive behavior (linking, shaking or biting the paw) was counted for 20 s.

*Tail suspension test*: For evaluation of depressive-like behavior, mice were suspended by the tail using adhesive tape 50 cm above the floor, for 6 min. Immobility time was defined as the amount of time during which the mice hung passively, without actively trying to release their tails from tape.

*Sucrose Splash Test (SST)*: Mice were placed under inverted glass funnels (14 cm diameter) and were allowed to habituate to the new environment for 3 min. Mice were then sprayed in the dorsum with ~0.7 ml of 10% (w/v) sucrose solution in water and behavior was monitored during the following 5 min. Latency to start grooming and cumulative self-grooming were recorded manually by a trained researched blind to the experimental conditions.

**Western blotting**. Bilateral hippocampi were dissected and homogenized in ice-cold RIPA buffer (pH 7.4, 50 mM Tris-HCl, 150 mM NaCl, 1% Triton X-100, 0.1% SDS, and 0.5% Na-deoxycholate [w/v] with protease and phosphatase cocktail inhibitors) and centrifuged at $1000 \times g$ for 10 min for removal of debris. Supernatants were stored at −80 °C until use. Protein concentration was determined by the Lowry method. Thirty milligram of protein per sample were resolved in a 10% SDS-PAGE gel run at a 20 mA constant current. Proteins were electroblotted at a 15 V constant voltage in a Trans-Blot® SD Semi-Dry Transfer Cell (Bio-Rad Labs, Inc.) to PVDF membranes (Bio-Rad Labs, Inc.). Membranes were blocked with 5% bovine serum albumin and incubated overnight with rabbit anti-synaptophysin (1:20,000, Abcam #32127), mouse anti-PSD95 (1:10,000, Abcam #2723), or rabbit anti-beta III tubulin (1:12,000, Abcam #18207) primary antibodies. Membranes were washed three times with TBS-T (pH 7.5, 50 mM Tris-Cl, 150 mM NaCl, 0.05% Tween-20), incubated for 1 h at room temperature with goat anti-rabbit secondary antibody conjugated to HRP (1:80,000, Abcam, #6721) and again washed three times with TBS-T. Proteins were visualized on a ChemiDoc XRS +(Bio-Rad) device using Super Signal West Pico Chemiluminescent Substrate (Thermo Scientific) following manufacturer's instructions.

**Electrophysiology in acute slices of mouse dorsal hippocampus**. Mock- or ZIKV-infected adult mice (6–10 days post-infection) were deeply anesthetized with isoflurane and intracardially perfused with a modified protective solution composed of (in mM) N-methyl-D-glucamine 100, HCl 90, NaHCO$_3$ 25, KCl 2.5, NaH$_2$PO$_4$ 1.25, MgSO$_4$ 10, CaCl$_2$ 0.5, glucose 25, HEPES 20, Na-pyruvate 3, N-acetylcysteine 2, ascorbic acid 0.5, which was chilled and oxygenated[58]. The brain was rapidly removed and 400 μm slices were cut in protective solution on a Vibratome at a plane 20° caudal-rostral and 10° ventral-dorsal off from sagittal, to preserve the main circuits of the dorsal hippocampus. The slices were transferred to a holding chamber containing protective solution at 32 °C saturated with 95% O$_2$–5% CO$_2$. During 30 min, the medium was gradually replaced (in 5 steps) with standard artificial cerebrospinal fluid (ACSF), composed of (in mM) NaCl 125, NaHCO$_3$ 25, KCl 2.5, NaH$_2$PO$_4$ 1.25, MgSO$_4$ 1.3, CaCl$_2$ 2.4, glucose 15, HEPES 5, at 32 °C. Slices were allowed to cool to room temperature and recover for 3–7 h before use. Individual slices were transferred to a custom 300 μl submersion chamber where they were suspended between nylon meshes and perfused with ACSF at 1.5 ml.min$^{-1}$. A glass microelectrode filled with ACSF (2–3 MΩ) was placed in CA3b stratum lucidum closer to stratum radiatum for field potential recording, and a bipolar insulated Pt stimulation electrode was placed near the proximal/hilar end of stratum lucidum. Two fEPSP (40 ms apart) were evoked every 20 s by 0.1-ms pulses delivered by a constant-current stimulator (DS3, Digitimer, UK). Stimulus intensity (0.06–0.25 mA) was adjusted to yield 30–40% of the fEPSP amplitude evoked by 1 mA (first pulse of the pair). Field potentials were recorded with an Axoclamp 2 A (Axon Instruments, USA) in bridge mode, amplified ×500, low-pass filtered at 3 kHz and digitized at 10 kHz under control of the LTP program[59]. After securing 20 min of stable fEPSP amplitude and slope, high-frequency stimulation (HFS) was applied as three trains, 10 s apart, of 100 pulses at 100 Hz, resuming low-frequency paired stimulation for 40 min thereafter. Three sweeps were averaged for each measurement of fEPSP slope and amplitude with WinLTP 2.30[59]. Slope was determined in the 2–4 ms initial portion of the first fEPSP after the pre-synaptic fiber volley, and data were expressed as percent of baseline, which was the mean of the measurements obtained 10 min before HFS. The baseline amplitudes of the second and first paired fEPSPs were ratioed to yield the paired-pulse ratio (PPR). LTP was expressed as the mean fEPSP slope measured 35–40 min post-HFS. Experimenters were blind to treatment.

**Flow cytometry**. Mouse brains without the cerebellum were coarsely chopped and incubated in a 0.01% collagenase (Sigma–Aldrich, #C0130) solution in PBS at 37 °C for 40 min under agitation. Samples were passed through a 70 μm filter in a total volume of 20 mL PBS. Cell suspension was mixed with a Percoll solution (30% Percoll, GE Healthcare #17089101, 3.33% PBS 10x, and 66,6% PBS 1 ×) and centrifuged for 30 min at $1500 \times g$. After centrifugation, the supernatant was discarded. Cells forming the pellet were washed twice with PBS, resuspended in PBS supplemented with 2% FCS and incubated for 30 min on ice with fluorescence-conjugated antibodies (1:100 dilution; anti-CD45 FITC, Tonbo Biosciences clone 30-F11; anti-CD11b PECy7, eBiosciences clone M1/70; anti-CD4 PE, Invitrogen #12-0042082; anti-CD8a APC, eBiosciences clone 53-6.7). Samples were washed twice with PBS before cytometry. Acquisition was performed on a MoFlo XDP FACS equipment (Beckman Coulter Inc., Brea, CA). Sorted cells were CD45$^{hi}$CD4$^+$ (T CD4 lymphocytes), CD45$^{hi}$CD8$^+$ (T CD8 lymphocytes), CD45$^{hi}$CD11b$^{hi}$ (activated microglia and/or infiltrating macrophages) and CD45$^{int}$CD11b$^{lo}$ (quiescent microglia). Data were analyzed with FlowJo vX software (Tree Star Inc., Ashland, OR).

**Illustrations**. Illustrations from images in Fig. 1a, Fig. 2a and Fig. 2f were created using *MindtheGraph* (www.mindthegraph.com; under FBA and CPF subscriptions) and subsequently modified (free culture Creative Commons license).

**Reporting summary**. Further information on research design is available in the Nature Research Reporting Summary linked to this article.

## Data availability

Data shown in Fig. 1b; Fig. 2b–e; Fig. 3a–f, i–k, n–t; Fig. 4a–e,o, q, r, t, u; Fig. 5c, g–n; Fig. 6d–e, i–p; Fig. 7a–f, k–n, Supplementary Fig. 1; Supplementary 2f; Supplementary Fig. 3; Supplementary Fig. 4; Supplementary Fig. 5c–g, j–l, o–q, t–v; Supplementary Fig. 6c, f, i; Supplementary Fig. 8c, f; Supplementary Fig. 9c, f; Supplementary Fig. 10 g–l; Supplementary Fig. 12; Supplementary Fig. 13 are provided as Source Data files. All other data are available from the corresponding authors upon reasonable request.

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

## Acknowledgements

We thank Melissa Florence, Jadilma Ferreira, Luis Fernando Fragoso, Bruno Maia da Silva Santos and Cecília Bataglioli Cavazzoni for technical support. Fundação de Amparo à Pesquisa do Estado do Rio de Janeiro (C.P.F., F.B.A., J.D.Z., M.B., J.M.S., N.G.C., S.V.A. L., F.G.D.F., I.A.M., J.R.C., A.T.P., and S.T.F.), Institutos Nacionais de Pesquisa—Instituto Nacional de Neurociência Translacional (F.G.D.F., S.T.F.), Conselho Nacional de Desenvolvimento Científico e Tecnológico (C.P.F., F.B.A., P.S.F., J.D.Z., M.B., J.M.S., N. G.C., S.V.A.L., F.G.D.F., I.A.M., J.R.C., A.T.P., and S.T.F.), Institutos Nacionais de Pesquisa—Inovação em Medicamentos e Identificação de Novos Alvos Terapêuticos (C.P. F.), Coordenação de Aperfeiçoamento de Pessoal de Nível Superior (R.L.S.N., C.S., I.N.S., and D.Z.) and Financiadora de Estudos e Projetos (A.T.P.).

## Author Contributions

C.P.F., F.B.A., R.L.S.N., H.A.P.N., F.G.D.F., I.A.M., J.R.C., A.T.P., and S.T.F. contributed to experimental design. C.P.F., F.B.A., R.L.S.N., H.A.P.N., P.S.F., J.D.Z., I.A.M., J.R.C., A. T.P., and S.T.F. analyzed and discussed data. C.P.F., F.B.A., F.G.D.F., I.A.M., J.R.C., A.T. P., and S.T.F. wrote the manuscript. R.L.S.N., J.D.Z., I.A.M., and A.T.P. performed viral replication. F.B.A., P.S.F., C.S., I.N.S., D.Z., and A.S.S. performed experiments in mice. Human adult cortical tissue was provided by J.M.S. and S.V.A.L. M.B. provided Il1r1 −/− mice and designed experiments in these mice. N.G.C and G.A.N designed, performed, and analyzed electrophysiological experiments. Histological and immunohistochemistry analyses were performed by C.P.F., F.B.A., P.S.F., V.L.S., and A.L.A.G. Molecular experiments were performed and analyzed by C.P.F., F.B.A., R.L.S.N., P.S.F., I. N.O.S., J.R.C., and I.A.M.

## Additional information

**Competing interests:** The authors declare no competing interests.

**Peer Review Information**: *Nature Communications* thanks the anonymous reviewers for their contribution to the peer review of this work. Peer reviewer reports are available.

