## [Peer Review File · Nature Communications]

Reviewers' comments:

Reviewer #1 (Remarks to the Author):

In recent reports, Zika virus (ZIKV) neuroinvasion in adults has been associated with neurologic sequelae including memory disturbances, which persist after viral clearance. The authors of the current study state they have developed a model of ZIKV encephalitis in adult mice to study the mechanism of this phenomenon. They show that intracerebroventricular inoculation of the Zika virus (ZIKV) Brazilian strain into adult Swiss mice leads to defects in novel object recognition only during acute encephalitis, when virus continues to replicate. Thus, the model they have developed does not actually model post-infectious neurologic sequelae associated with ZIKV encephalitis in adults. Using their model, the investigators convincingly demonstrate that ZIKV targets neurons throughout the central nervous system (CNS) with high levels detected in hippocampal and striatal tissues. They also show that synaptophysin staining is diminished, mRNA levels of C1q are increased, and mice perform poorly on memory tests during time-points when ZIKV is actively replicating within the CNS, but not after viral clearance. Treatment with anti-TNF antibody or minocycline prevents memory dysfunction and is associated with improvement in levels of synaptophysin and C1q. The investigators also show that IBA-1 staining in brain sections from mice with actively replicating virus is elevated on cells that are ameboid or stellate in appearance. The authors conclude that ZIKV infection promotes microglial-mediated synapse loss via a mechanism that involved complement and/or TNF, however, there are no data provided that specifically demonstrate this mechanism. In addition, evaluation of neuroinflammation and its effects on the CNS during ZIKV encephalitis is incomplete, as the authors do not characterize the extent and types of infiltrating immune cells, cellular sources of cytokines, or distinguish macrophages from activated microglia. They also do not convincingly determine whether there is significant neuronal death or dysfunction. Overall, this is an intriguing, but mostly correlative and therefore preliminary, study that shows a role for TNF in memory disturbances during acute ZIKV encephalitis, via an unknown mechanism and using a model which is of unclear significance with regard to our understanding of memory disturbances that occur in patients that have recovered from ZIKV neuroinvasive disease.

Specific concerns:

Figure 1: The investigators utilized temporal lobe tissues from unspecified cortical regions from patients undergoing surgery for epilepsy and show that ZIKV replicates in these tissues over a four-day period. No information is provided regarding the specific cortical regions evaluated and the cellular targets are not determined. This information is important because of the variability in replication levels between samples and for determining the relevance of this data for the murine findings. In this Figure, the investigators also show detection of virus NS2B via immunostaining within the CA1 and dentate gyrus (DG) regions of the hippocampus and in the striatum. Given the lack of co-staining for NeuN in some of the NS2B+ cells and prior reports demonstrating tropism of

ZIKV for neural precursor cells (NPC), the investigators should utilize NPC markers to determine other possible targets for ZIKV.

Figure 2: While the investigators show memory impairment during time-points when virus is replicating, the basis for memory impairment is poorly evaluated. The investigators show low power images for synaptophysin staining using standard immunohistochemistry of the CA1 and DG regions of the hippocampus in which there appears to be less DAB staining on the whole section. This is not satisfactory for convincingly demonstrating loss of synapses, which requires much higher power techniques (confocal, super-resolution microscopy, etc) and functional analyses. In addition, learning and memory in mice has been traced to mossy fiber synapses within the CA3 regions, which has not been evaluated. Finally, given that memory deficits are temporary, the authors should demonstrate whether synapses are repaired and whether they are then functional.

Figure 3: The authors use IBA-1 immunostaining to evaluate microglia. However, this marker is also expressed by infiltrating macrophages, which are not evaluated. This is important, because infiltrating immune cells are also cellular sources of cytokines such as TNF.

Figure 4: The data in this figure are intriguing, as it suggests that targeting TNF via use of neutralizing antibodies or minocycline, which inhibits TNF-mediated apoptosis, might prevent memory dysfunction. The authors also show that minocycline prevents loss of immunostaining for synaptophysin and decreases levels of C1q mRNA during acute ZIKV infection. Again, the synaptophysin analyses are unconvincing and the source of complement is unclear, especially given the lack of continued C1q expression after amelioration of encephalitis. It is possible that acute encephalitis, which is often associated with disruption of blood-brain barrier integrity, might lead to complement penetration of the CNS from peripheral sources, including immune cells. Thus, while TNF appears to play a role in memory dysfunction during acute ZIKV encephalitis, the mechanism is unclear.

Reviewer #2 (Remarks to the Author):

In this study, Figueiredo and colleagues explore the functional consequences of Zika virus (ZIKV) infection in the adult brain. While considerable focus has been placed on the terrible consequences of ZIKV infection in children born from mothers that were infected while pregnant, ZIKV infection of adults can have neurological consequences (albeit infrequently). Here, the authors use to models to establish the capacity of ZIKV to replicate in cells of the brain, demonstrate that infection of CNS

tissues impacts cognition via the involvement of the activation of innate and inflammatory factors.

The paper is certainly of interest as cognitive and behavioral consequences of ZIKV infection may be a yet to be fully appreciated public health problem among the large number of individuals infected during the recent epidemic.

Overall, the manuscript is well-written and describes some nice experiments. My principal (somewhat substantial) concern with the manuscript relates to the murine challenge model used throughout. In the author's experiments, they use an intracranial inoculation method to deliver a rather large amount of virus into this compartment and then observe viral tropism and replication. There is no guarantee that this biology accurately reflects circumstances that occur during/following neuroinvasion, which has the potential to deliver virus in different locations, via different mechanisms (free virus/cell associated), in much different doses, and finally in a different immunological/inflammatory context. While the authors do demonstrate that delivery into two different portions of the brain yield equivalent results, this model remains a "what is possible" system, rather than what occurs. I recognize the challenges of doing experiments in adult immunologically competent animals inoculated via a peripheral route, but at least some bridging experiments are required to connect this neurovirulence model to what happens in nature. I feel this is critical for publication.

The composition of the display items requires adjustment. The large number of panels in each of these display items reduces clarity. For example, the human brain studies in Fig 1a-d should be a different figure.

Minor points:

The authors do not provide any context for their study in an epidemiological sense. They reference examples of neurological complications in adults, but do not provide data detailing how frequently these outcomes have been observed. While even rare outcomes merit careful mechanistic study, the size of this problem will not be clear to the reader.

Reviewer #3 (Remarks to the Author):

Figueiredo CP et al showed that after intracerebroventricular injection of ZIKV into adult mice, neurons but not microglia or astrocytes in the frontal cortex and hippocampus were preferentially targeted. These mice showed presynaptic terminal damage and memory deficits that lasted for 30 days after infection. The memory deficit spontaneously recovered 60 days after infection. However, no motor-related defects were observed in these mice, despite that ZIKV was identified in striatum, cerebellum, spinal cord as well as peripheral nerves such as sciatic nerve. They also provided evidence that neutralizing inflammation or blockage of microglial polarization rescue the memory deficits. The results of this study are potentially interesting that adds to the literature of virus-induced memory impairment. However, some of the results need clarification to better understand the mechanism.

1. A study using adult mouse model showed that ZIKV infects neural progenitors in the forebrain and hippocampus (Li et al Cell Stem Cell 2016). On the other hand, the authors showed that ZIKV infects the CNS, PNS as well as non-neural tissue in adult mice, which is more compatible with some of the clinical reports. As far as I can tell, the difference in these two studies are 1) IRF mice vs Swiss mice; 2) 10^5 PFU vs 10^3 FFU; 3) retro-orbitally vs intracerebroventricular injection; 4) Asian vs American strain of ZIKV. Is it possible that the higher titer of the virus as well as direct injection of the virus into the brain induced a more widespread infection? Or are there any other factors that may play a role here? The differences between these two studies are puzzling and need some clarification.

2. The authors found that Zika virus concentration was much higher in frontal cortex, hippocampus and striatum comparing to other CNS and PNS tissue. What is intriguing is that there is almost no immunoreactivity of ZIKV NS2B in the thalamus and subventricular zone. What makes frontal cortex and hippocampus more susceptible to ZIKV infection? NS4A/B inhibit Akt-mTOR signaling in human fetal neural stem cells (Liang Q et al Cell Stem Cell 2016). Is Akt-mTOR or other related signaling pathways responsible for the preference in infection?

3. The authors showed that ZIKV infection targeted neurons but not astrocytes and microglial. Do the infected neurons trigger inflammation and microgliosis? What is the possible cause of inflammation and microgliosis?

4. Two puzzling results from the behavioral tests: 1) despite similar concentration of virus as well as NS2B expression in the striatum and hippocampus, the infected mice did not exhibit locomotor and exploration deficits; 2) despite similar concentration of virus as well as NS2B expression in the frontal cortex and hippocampus, the infected mice did not exhibit anxiety deficits. It would be helpful to show whether synapse damage, brain inflammation and microgliosis occur in the striatum and frontal cortex as well.

5. In the novel object recognition test, the variability in exploration time is much larger at 30 dpi in comparison to 1 dpi and 14 dpi. A paired t-test will be helpful to better illustrate the exploration of familiar and novel object for each mouse at different time points.

6. What is the virus concentration or NS2B signal in other cortical areas such as motor, somatosensory or entorhinal cortices? This result may provide some evidences for memory but not motor or sensory deficits.

7. The authors showed a loss of pre-synaptic terminals in CA1 and dentate gyrus of ZIKV-infected adult mice. Majority of synaptic input into CA1 and dentate gyrus are from entorhinal cortex (located in the medial temporal lobe), with some local input from CA3. Therefore, part of the presynaptic loss in CA1 and dentate gyrus is not from hippocampus but cortex. In order to clarify which synapses go wrong in the hippocampus, it would be helpful to show pre-synaptic (Synaptophysin) as well as postsynaptic (Homer1) puncta using Synaptophysin and Homer 1 immunostaining in CA1 and dentate gyrus. It would be even better to show the data in CA3 as well.

8. Memory deficits lasted for at least 30 days after infection and recovered at 60 dpi. However, the authors provided data for synapse damage, brain inflammation and microliosis only at 6 dpi. To demonstrate a close correlation between the memory deficits and synaptic/inflammation changes, the authors are suggested to show the same data at 30 and 60 dpi, respectively.

9. The authors showed the rescue of memory deficits using Infliximab, a TNF- α antibody. It is not clear whether the inflammation directly leads to memory deficits or indirectly causes memory deficits through synapse damage. It will be helpful to show the immune data of synaptophysin as well as Homer1 in the Infliximab-rescued mice.

Minor comments:

1. In page 6, the authors claim that Fig. 1j showed staining of pyramidal and granule cells in the hippocampus. How did the authors identify that the cells are pyramidal and granule cells?

2. In page 6-7, what is the brain region for Fig. 1 t-j'?

3. In page 10, Suppl. Fig. 3h-k should be Suppl. Fig. 2h-k.

4. In page 12, n = 7 - 10 mice/group for Fig. 2d. Please specify the exact number of mice in each group.

5. In page 19, (triple deficient in interferon regulatory factor)²⁵. This reference should be reference 9.

6. In page 40-41, references 7 and 12 are the same.

RESPONSE TO REFEREES LETTER

Reviewer #1

1) “They show that intracerebroventricular inoculation of the Zika virus (ZIKV) Brazilian strain into adult Swiss mice leads to defects in novel object recognition only during acute encephalitis, when virus continues to replicate. Thus, the model they have developed does not actually model post-infectious neurologic sequelae associated with ZIKV encephalitis in adults.”

-- We thank the reviewer for bringing up this important point. To address this issue, we have investigated whether memory impairment was restricted to acute stages of infection and viral replication, or whether it persisted after resolution of the infection. We performed qPCR experiments in the brains of mice at 6, 30 or 60 days post infection (dpi) to determine viral negative RNA strand (which is required for viral replication). As shown in new Suppl. Fig. 1c, we found that viral replication was high at 6 dpi, and levels of negative strand ZIKV RNA approached undetectable levels at 30 and 60 dpi.

We further note that ZIKV-infected mice exhibit memory impairment between 1 and 30 dpi (Fig. 3a-c, e), with memory recovery observed at 60 dpi (Fig. 3d, f). Thus, results at 30 dpi indicate that memory impairment persists for some time after resolution of the acute infection/viral replication phase in mice. Accordingly, we have rephrased our description of these results in the revised manuscript to state that memory deficits persist for some time following infection resolution but are not permanent in ZIKV-infected mice.

2) “The authors conclude that ZIKV infection promotes microglial-mediated synapse loss via a mechanism that involved complement and/or TNF, however, there are no data provided that specifically demonstrate this mechanism.”

-- We thank the reviewer for the opportunity to address in more detail the interplay between TNF α , complement system proteins and synapse damage in ZIKV-infected mice. We have performed additional experiments aimed to strengthen the link between these events. In the original version of the manuscript, we showed that

treatment with minocycline, a drug that inhibits microglial polarization to a proinflammatory phenotype, ameliorated ZIKV-induced memory impairment (Fig. 5a in the revised manuscript). Interestingly, however, we found that minocycline treatment did not block the increase in TNF α expression induced by ZIKV infection, suggesting that activated macrophages or microglia (see below) are not the main source of TNF α in ZIKV-infected mice (New Fig 5n).

We further show that protection against ZIKV-instigated memory impairment by the TNF α -neutralizing antibody infliximab (reported in the original manuscript; Fig. 5b in the revised ms) is accompanied by protection against hippocampal synapse loss (New Fig. 5c-h). In addition, we show that infliximab blocks hippocampal microglial activation in ZIKV-infected mice (New Fig. 5i-m). These findings indicate that TNF α triggers microglial activation and synapse damage in mice.

Finally, we investigated in more detail the role of complement proteins in synapse damage in ZIKV-infected mice. A previous study showed that aberrant activation of complement in response to brain inflammation caused by West Nile Virus infection promotes microglia-mediated synapse loss (Vasek et al., 2016, Nature). As reported in our original manuscript, ZIKV-infected animals presented up-regulated expression of C1q and C3 (Fig. 6a,b in the revised manuscript). Under physiological conditions, complement proteins in the brain are largely produced locally, mainly by microglia, since the blood–brain barrier protects the CNS from peripheral complement and immune cells. In neurodegenerative disorders, microglia have been implicated as the main source of complement that drives synapse loss. However, neurons and astrocytes also express C1q and C3 proteins (Litvinchuk et al., 2018, Neuron; Stevens et al., 2007, Cell; for a review, see Presumey et al., 2017, Adv in Immunol.). In the revised manuscript, we demonstrate that treatment with minocycline (an inhibitor of microglial polarization) but not with infliximab (a TNF α -neutralizing antibody) decreased brain expression of C1q induced by ZIKV, and both treatments attenuated the up regulation of C3 expression in the brains of ZIKV-infected mice (new Fig. 6c-f). The lack of effect of infliximab on C1q expression in ZIKV-infected mice may be, at least in part, explained by neuronal production of C1q (Stevens et al., 2007, Cell; Presumey et al., 2017, Adv in Immunol.). Our results

further suggest that both resident and infiltrating peripheral immune cells may be relevant sources of C3 in ZIKV-infected mice, since both central (i.c.v., Fig. 6f) and peripheral (i.p., Suppl. Fig. 11b) treatment with infliximab decreased brain expression of C3.

We further show that i.c.v. infusion of either anti-C1q or anti-C3 antibodies preserves synapses (new Fig. 6g-m) and prevents memory impairment (new Fig. 6n) in ZIKV-infected mice. Collectively, results suggest that ZIKV infection leads to increased production/release of TNF- α , microglial polarization and increased expression of complement system proteins, leading to synapse damage and memory impairment. Inhibition of microglial polarization, blockade of TNF- α or neutralization of complement system proteins prevent ZIKV-induced synapse loss and memory deficit.

3) “In addition, evaluation of neuroinflammation and its effects on the CNS during ZIKV encephalitis is incomplete, as the authors do not characterize the extent and types of infiltrating immune cells, cellular sources of cytokines, or distinguish macrophages from activated microglia.”

-- We agree with the reviewer that this important aspect of the work was missing in the original manuscript, and we thank them for the opportunity to include a more detailed investigation of neuroinflammation triggered by ZIKV infection in our study. We first note that, in the original version of the manuscript, we had included results on brain immunoreactivity of TMEM119, a specific microglial marker, in ZIKV-infected mice (Fig. 4k-o). Those results suggested that resident microglia undergo activation following viral infection. To determine whether/which peripheral immune cell types infiltrate the brain following infection, we performed fluorescence activated cell sorting (FACS) analyses at the peak of viral replication. We found that brains of ZIKV-infected mice showed increased infiltrating CD4+ (Helper) and CD8+ (cytotoxic) T lymphocytes (Fig. 4s-u).

We further used CD45 and CD11b staining as a strategy to study myeloid cell dynamics in the brains of ZIKV-infected mice, since there is currently no consensus in the literature regarding an approach to distinguish activated microglia from infiltrating macrophages (Koeniger & Kuerten, 2017, Int J Mol Sci.; Greter et al.,

2015, *Front Immunol.*). This approach allowed us to distinguish between resting microglia (CD45^{int}/CD11b+) and activated myeloid cells (CD45^{hi}/CD11b+, comprising activated microglia and/or infiltrating macrophages). We found no differences in numbers of CD11b+CD45^{int} cells (resting microglia) (New Fig. 4p,q). On the other hand, ZIKV infection led to increased brain numbers of CD11b+CD45^{high} cells (New Fig. 4p,r). CD11b+CD45^{high} cells comprise a mixed population of infiltrating macrophages and activated microglia (Greter et al., 2015, *Front. Immunol.*). Since we found both significant microgliosis and leukocyte infiltration in ZIKV-infected brains, it is likely that this CD11b+CD45^{high} population is comprised of both resident and infiltrating myeloid cells. Collectively, our new results demonstrate that ZIKV-infected brains present increased numbers of T cells and activated myeloid cells, which are likely of both resident and peripheral origin.

4) They also do not convincingly determine whether there is significant neuronal death or dysfunction.

-- To address this issue, we performed FluoroJade B (FJ) staining of mouse brain sections 10 days post infection (dpi) by ZIKV. We found few cells positive for FJ staining in the CA1 region of the hippocampi (but not in the DG, CA3 or parietal cortex) of two out of five mice used in the experiment. FJ-positive cells were not found in the hippocampi or parietal cortex of additional five mice at 6 dpi. These results have been included in new Suppl. Fig. 6.

Specific concerns:

“Figure 1: The investigators utilized temporal lobe tissues from unspecified cortical regions from patients undergoing surgery for epilepsy and show that ZIKV replicates in these tissues over a four-day period. No information is provided regarding the specific cortical regions evaluated and the cellular targets are not determined. This information is important because of the variability in replication levels between samples and for determining the relevance of this data for the murine findings.”

-- Surgeries followed the international standard for anterior lobectomy, consisting of removal of tissue 3-4 cm from the pole of the temporal lobe, and concomitant

removal of the hippocampus (where epileptic foci were located). Samples used in the current study comprised temporal lobe cortical fragments plus associated subcortical tissue. Information in this regard has been included in "Methods" in the revised manuscript.

"In this Figure, the investigators also show detection of virus NS2B via immunostaining within the CA1 and dentate gyrus (DG) regions of the hippocampus and in the striatum. Given the lack of co-staining for NeuN in some of the NS2B+ cells and prior reports demonstrating tropism of ZIKV for neural precursor cells (NPC), the investigators should utilize NPC markers to determine other possible targets for ZIKV."

-- As suggested, we performed additional experiments to determine whether neural precursor cells were targeted by ZIKV in the brains of adult immunocompetent mice used in our study, since previous reports showed that NPCs are the main cell type targeted by the virus in immunocompromised mice. We found no NS2B immunostaining in doublecortin-positive cells at 6 dpi (new Suppl. Fig. 2a-c). However, we found that ZIKV infection decreased the number of doublecortin-positive cells in the hippocampal dentate gyrus, compared to mock-injected mice (new Suppl. Fig. 2d-f). It is, thus, possible that ZIKV targets NPCs at early stages of infection, leading to cell death and to lack of detection of NS2B immunostaining in NPCs in the hippocampus of adult immunocompetent mice.

"Figure 2: While the investigators show memory impairment during time-points when virus is replicating, the basis for memory impairment is poorly evaluated. The investigators show low power images for synaptophysin staining using standard immunohistochemistry of the CA1 and DG regions of the hippocampus in which there appears to be less DAB staining on the whole section. This is not satisfactory for convincingly demonstrating loss of synapses, which requires much higher power techniques (confocal, super-resolution microscopy, etc) and functional analyses. In addition, learning and memory in mice has been traced to mossy fiber synapses within the CA3 regions, which has not been evaluated. Finally, given that memory

deficits are temporary, the authors should demonstrate whether synapses are repaired”

-- As requested, we have performed double immunofluorescence labeling of synapses using pre- and post-synaptic markers (synaptophysin and Homer1, respectively, as suggested by reviewer #3) followed by confocal imaging in brain sections from ZIKV-infected mice at 6 and 60 dpi (new Fig. 3g-p). At the peak of viral replication (6 dpi), we found that the CA3 hippocampal subregion of ZIKV-infected mice presented decreased levels of pre-synaptic (new Fig. 3g-i), but not post-synaptic (new Fig. 3g,h,j) puncta, resulting in reduced co-localized synaptic puncta (new Fig. 3g,h,k), when compared with mock-infected mice at 6 dpi.

Interestingly, synaptic damage induced by ZIKV was reversible, since the density of both pre-synaptic and co-localized synaptic puncta returned to control, mock-infected levels when evaluated at 60 dpi (Fig. 3l-p). These findings corroborate results from Western blotting analyzes of hippocampal tissue from ZIKV-infected mice for pre- (synaptophysin) and post-synaptic (PSD-95) proteins (Suppl. Fig. 4f,g in the revised manuscript). No changes in density of pre-synaptic or co-localized synaptic puncta were detected in the CA1 hippocampal subregion in mock- versus ZIKV-infected animals (new Suppl. Fig. 4h-l).

Following the reviewer’s suggestion, we also investigated the impact of ZIKV infection on synaptic plasticity (long term potentiation, LTP) by performing electrophysiological recordings of fEPSPs evoked through the mossy fiber pathway in dorsal hippocampal slices. Results showed that both post-tetanic and long-term potentiation of mossy fiber synapses in the dorsal hippocampus were inhibited in slices from ZIKV-infected mice (new Fig. 3q-s). Moreover, and consistent with a predominantly pre-synaptic impact of ZIKV (revealed by decreased synaptophysin immunoreactivity; Fig. 3g-i), paired-pulse ratio was also markedly inhibited in slices from ZIKV-infected mice (new Fig. 3t).

“Figure 3: The authors use IBA-1 immunostaining to evaluate microglia. However, this marker is also expressed by infiltrating macrophages, which are not evaluated. This is important, because infiltrating immune cells are also cellular sources of cytokines such as TNF.”

-- We initially note that, in the original manuscript, we had included results on brain immunoreactivity of TMEM119, a specific microglia marker, in ZIKV-infected mice (Fig. 4k-o). Those results suggested that resident microglia undergo activation following viral infection. Distinction between activated microglia and infiltrating macrophages is complicated as both cell populations, upon activation, present similar expression of various cell markers, none of which can be considered exclusive of one particular population (Koeniger & Kuerten, 2017, *Int J Mol Sci.*; Greter et al., 2015, *Front Immunol.*). To address in more detail whether/which peripheral immune cell types infiltrate the brain following infection, we performed fluorescence activated cell sorting (FACS) analyses at the peak of viral replication (6 dpi). We found that ZIKV infection did not affect brain numbers of resident quiescent microglial cells (CD11b+CD45^{int} cells) but increased the numbers of activated myeloid cells (CD11b+CD45^{high}) which, by this approach, cannot be distinguished between activated microglia and infiltrating peripheral macrophages (Fig. 4p-r). Moreover, brains of infected mice showed increased infiltrating CD4+ (Helper) and CD8+ (cytotoxic) T lymphocytes (Fig. 4s-u).

Collectively, these results establish that ZIKV infection leads to brain infiltration by T lymphocytes and to increased numbers of activated myeloid cells.

“Figure 4: The data in this figure are intriguing, as it suggests that targeting TNF via use of neutralizing antibodies or minocycline, which inhibits TNF-mediated apoptosis, might prevent memory dysfunction. The authors also show that minocycline prevents loss of immunostaining for synaptophysin and decreases levels of C1q mRNA during acute ZIKV infection. Again, the synaptophysin analyses are unconvincing and the source of complement is unclear, especially given the lack of continued C1q expression after amelioration of encephalitis. It is possible that acute encephalitis, which is often associated with disruption of blood-brain barrier integrity, might lead to complement penetration of the CNS from peripheral sources, including immune cells. Thus, while TNF appears to play a role in memory dysfunction during acute ZIKV encephalitis, the mechanism is unclear.

-- We thank the reviewer for the opportunity to address in more detail the interplay between TNF α , complement system proteins and synapse damage in ZIKV-infected

mice. We have performed additional experiments, which strengthen the link between these molecular events. In the original manuscript, we showed that the expression of complement proteins (C1q and C3) was up-regulated in the brains of ZIKV-infected mice (Fig. 6a,b in the revised ms), and that minocycline (an inhibitor of microglial polarization) or infliximab (a TNF-neutralizing antibody) ameliorated ZIKV-induced memory impairment (Fig. 5a,b in the revised ms). We now show that protection against ZIKV-instigated memory impairment by infliximab (reported in the original manuscript) is accompanied by decreased hippocampal microgliosis (new Fig. 5i-m) and protection against hippocampal synapse loss (New Fig. 5c-h). Interestingly, we also found that minocycline treatment did not block the up-regulation of TNF- α induced by ZIKV infection (New Fig. 5n).

Further implicating the TNF- α -complement-microglia axis in ZIKV-instigated synapse damage and memory deficits, we now show that both minocycline and infliximab treatments resulted in reduced expression of C3 in the brains of ZIKV-infected mice (New Fig. 6d, f). While we agree with the reviewer that the high levels of complement proteins in the CNS of ZIKV-infected mice could be due to disruption of BBB integrity and penetration of peripheral complement, our finding that the expression of C3 (and C1q) in the brains of infected mice was elevated implies local production in the brain. We further note that C3 could originate from activated resident microglial cells or infiltrating macrophages, since both central (i.c.v, Fig. 6f) and peripheral (i.p., Suppl. Fig. 11b) treatment with infliximab reduced brain expression of C3. As noted above, the lack of effect of infliximab on brain expression of C1q in ZIKV-infected mice may be related to the capacity of neurons to produce C1q (Stevens et al., Cell, 2007; For review see Presumey et al., 2017, Adv Immunol.).

As also noted above, and as requested by the reviewer, our new analyses of synapse density have been performed using double immunofluorescence labeling of pre- and post-synaptic markers (synaptophysin and Homer1, respectively) followed by confocal imaging and analysis (new Fig. 3g-p, Fig. 5c-h). In the revised manuscript, we additionally show that infliximab decreases microglial activation (New Fig. 5i-m), and minocycline treatment does not reduce TNF- α upregulation, suggesting that TNF- α is a key molecule inducing microglial activation and synapse loss in ZIKV-infected mice.

Finally, we now show that *i.c.v.* infusion of either anti-C1q or anti-C3 antibodies in ZIKV-infected mice preserves synapses (Fig. 6g-m) and prevents memory impairment (Fig. 6n). Taken together, original and new results support the notion that ZIKV induces increased production of TNF- α , microglial activation and up-regulation of complement system proteins, thus leading to a deleterious impact on synapses and memory.

Reviewer #2

1) “My principal (somewhat substantial) concern with the manuscript relates to the murine challenge model used throughout. In the author’s experiments, they use an intracranial inoculation method to deliver a rather large amount of virus into this compartment and then observe viral tropism and replication. There is no guarantee that this biology accurately reflects circumstances that occur during/following neuroinvasion, which has the potential to deliver virus in different locations, via different mechanisms (free virus/cell associated), in much different doses, and finally in a different immunological/inflammatory context. While the authors do demonstrate that delivery into two different portions of the brain yield equivalent results, this model remains a “what is possible” system, rather than what occurs. I recognize the challenges of doing experiments in adult immunologically competent animals inoculated via a peripheral route, but at least some bridging experiments are required to connect this neurovirulence model to what happens in nature. I feel this is critical for publication.”

-- We thank the reviewer for raising this important point, which we have addressed by performing a number of additional experiments in the revised manuscript.

First, we investigated viral distribution/replication in different brain regions following infection in immunocompromised animals (A129 mice, deficient in type I interferon receptor), a model frequently used in studies that investigated the neurological consequences of systemic ZIKV infection. As might be expected, we found that, following intravenous (*i.v.*) infection (retro-orbital injection), ZIKV reaches the brains of A129 mice. Interestingly, however, high and quite similar viral titers were observed throughout the brain in such mice (Suppl. Fig. 1g), in contrast with the regional selectivity for viral replication verified using *i.c.v.* infection in wild-type mice

(Fig. 2d and Suppl. Fig. 1d). Moreover, similar high titers were found in the central nervous system and in peripheral organs of ZIKV-infected A129 mice (Suppl. Fig. 1g). It appears likely, therefore, that lack of ability to mount an effective interferon-based immunological response leads to generalized replication of ZIKV in the entire brain, abolishing the preferential targeting of regions associated with cognitive and motor control verified in immunocompetent mice.

Second, we evaluated viral distribution in different brain regions following systemic (s.c.) infection by ZIKV in wild-type mice at post-natal day 3, an age at which WT immunocompetent mice are susceptible to ZIKV infection. Results were similar to those obtained with A129 mice, with high and similar viral load observed in all assessed brain regions (Suppl. Fig. 1h). Indeed, recent studies have shown that brains from newborn immunocompetent mice are susceptible to ZIKV invasion and replication (Nem et al., 2018, Sci Trans Med.; Li et al., 2018, Viruses), since an efficient interferon-mediated immune response appears to be lacking at this early stage of development.

Third, we assessed the ability of ZIKV to reach memory-related brain regions following retro-orbital injection (i.v.) in WT adult mice. Results showed that ZIKV RNA was detected (albeit at lower levels than following i.c.v. infection) in both frontal cortex (FC) and hippocampus (HPc), establishing that ZIKV targets memory-related brain regions when it reaches the brain through a peripheral route (Suppl. Fig. 1i). The reasons for the lower replication efficiency of ZIKV reaching the brain via a peripheral route (compared to i.c.v. inoculation) are still unclear, and we feel future studies are warranted to explore this feature of infection.

These new findings demonstrate that ZIKV targets memory-related brain regions regardless of whether the route of infection is central (i.c.v.) or peripheral. Interestingly, the fact that ZIKV replicates in high titers throughout the brain in immunocompromised (IFN response-deficient) mice suggests that the brain region-specificity of the host immune response may be an important determinant for the efficiency of viral replication in different brain structures. These findings reinforce the translational potential of our adult ZIKV infection model, since the virus has indeed been detected in CSF of patients presenting with neurological complications

(Carteaux et al., 2016, N Engl J Med.; Roze B, et al., 2016, Eur Surveill.; Azevedo et al., 2016, J Clin Virol.)

Finally, we asked whether the pattern of brain distribution of viral replication was dependent on the total amount of virus inoculated into the lateral cerebral ventricle of mice. To address this issue, we performed additional experiments with i.c.v. infection using a lower titer of ZIKV infectious particles (10^3 PFU, 100-fold lower than the 10^5 PFU used throughout our study). We found that preferential replication of the virus in memory-related brain structures persists with the lower titer (Suppl. Fig. 1f), suggesting that the brain region-specificity of viral replication is not dependent on the total amount of ZIKV introduced via i.c.v.

2) The composition of the display items requires adjustment. The large number of panels in each of these display items reduces clarity. For example, the human brain studies in Fig 1a-d should be a different figure.

-- We agree with the reviewer and have separated the human brain studies (new Fig. 1) from studies in rodents

3) Minor points:

The authors do not provide any context for their study in an epidemiological sense. They reference examples of neurological complications in adults, but do not provide data detailing how frequently these outcomes have been observed. While even rare outcomes merit careful mechanistic study, the size of this problem will not be clear to the reader.

-- We thank the reviewer for raising this issue. We have added a few sentences in the revised manuscript to clarify this point (page 3, li 69-72; page 17, li 357-58).

Reviewer #3

1) “However, no motor-related defects were observed in these mice, despite that ZIKV was identified in striatum, cerebellum, spinal cord as well as peripheral nerves such as sciatic nerve.”

-- We thank the reviewer for raising this interesting point, as this prompted us to investigate in more detail possible motor deficits in ZIKV-infected mice. Although our original observations indicated lack of exploratory deficits in ZIKV-infected mice in the open field task (Suppl. Fig. 3a,b in the revised ms), new experimental results revealed impaired performance in the rotarod test 5 days post infection (dpi) (new Suppl. Fig. 8h). Performance in the rotarod test returned to normal 30 dpi (new Suppl. Fig. 8i). Interestingly, we found microgliosis in the striatum at 6 dpi, but not 30 or 60 days after infection (new Suppl. Fig. 8a-g). Results suggest that increased striatal microgliosis is associated with poor motor performance in ZIKV-infected mice during the acute phase of infection.

2) “A study using adult mouse model showed that ZIKV infects neural progenitors in the forebrain and hippocampus (Li et al Cell Stem Cell 2016). On the other hand, the authors showed that ZIKV infects the CNS, PNS as well as non-neural tissue in adult mice, which is more compatible with some of the clinical reports. As far as I can tell, the difference in these two studies are 1) IRF mice vs Swiss mice; 2) 10^5 PFU vs 10^3 FFU; 3) retro-orbitally vs intracerebroventricular injection; 4) Asian vs American strain of ZIKV. Is it possible that the higher titer of the virus as well as direct injection of the virus into the brain induced a more widespread infection? Or are there any other factors that may play a role here? The differences between these two studies are puzzling and need some clarification.

-- We thank the reviewer for bringing up this important issue. As also noted in our response to reviewer # 2 (above), previous studies investigating the neurological consequences of ZIKV infection have used immunocompromised animals. Specifically, the study by Li and colleagues cited by the reviewer showed that neural precursors are the main cell type targeted by ZIKV in mice triply deficient in interferon regulatory

factor (IRF). Because (1) we have used immunocompetent adult mice, and (2) ZIKV has been detected in both CNS and CSF of infected patients, thus establishing that the virus does reach the CNS following infection in humans, we feel our approach may be advantageous to investigate the impact of ZIKV in the immunocompetent brain. Nonetheless, and as also requested by reviewer # 2, we have performed additional bridging experiments to clarify the similarities and differences in results obtained with the two alternative models, i.e., retro-orbital infection in IFN deficient mice versus i.c.v. infection in immunocompetent mice:

(i) We initially asked whether a similar pattern of viral distribution in different brain regions would be obtained following i.c.v. infection in WT mice or intravenous (i.e., retro-orbital) inoculation of ZIKV in Interferon receptor 1 knockout mice (A129 mice). We confirmed that the virus reaches and replicates in the brain in IFN receptor KO mice. Significantly, however, in the absence of interferon response, high viral loads were seen in all brain regions investigated (Suppl. Fig. 1g), in sharp contrast with the brain regional distribution pattern found following i.c.v. infection. This new result suggests that the ability to mount an effective interferon-mediated immune response may differ amongst brain regions and may underlie the different susceptibilities of distinct brain regions to infection by ZIKV (Fig. 2d, Suppl. Fig. 1d).

(ii) Because studies have shown that brains of newborn wild-type mice are susceptible to ZIKV invasion and replication, likely reflecting lack of an efficient interferon-mediated immune response at this early stage of development, we investigated brain infection following peripheral (subcutaneous) injection of ZIKV (10^6 PFU) in mice at post-natal day 3. Similar to the results described above with IFN receptor-deficient mice (Suppl. Fig. 1g), we found that viral mRNA levels were high and similar in all assessed brain regions in neonatally infected mice (Suppl. Fig 1h).

These results suggest that, while useful for other types of studies, immunodeficient mouse models may not be ideally suited to investigate the brain impact of ZIKV and, in particular, the differential vulnerabilities and structural/functional damage of distinct brain regions/structures following infection. For such studies, we feel our i.c.v. infection approach in immunocompetent adult mice represents an advantageous and more translational approach, since clinical studies have established that ZIKV indeed reaches the CNS following infection in

humans (Carteaux et al., 2016, *N Engl J Med.*; Roze B, et al., 2016, *Eur Surveill.*; Azevedo et al., 2016, *J Clin Virol.*).

(iii) We further investigated whether the brain regional distribution of ZIKV load was dependent on the amount of virus used for infection. To this end, we infected mice i.c.v. with a lower titer of ZIKV (10^3 PFU, compared to 10^5 PFU used in all other experiments in our study). Results showed that the regional distribution of viral replication was similar using both viral amounts (10^5 PFU, Fig. 2d) (10^3 PFU, Suppl. Fig. 1f). We found that, independent of the viral load injected in immunocompetent mice, ZIKV preferentially targets structures related to memory and motor control, suggesting that these brain structures are specially susceptible to infection (Suppl. Fig. 1e).

(iv) Finally, we performed intravenous (retro-orbital) infection in immunocompetent mice to determine whether ZIKV would reach the brain under these conditions. Interestingly, ZIKV mRNA was detected, albeit at lower levels, in both hippocampus and frontal cortex of mice following intravenous infection (10^6 PFU ZIKV) (Suppl. Fig. 1i). These results indicate that ZIKV targets memory-related brain regions when the virus reaches the brain via a peripheral (intravenous) route.

3) “The authors found that Zika virus concentration was much higher in frontal cortex, hippocampus and striatum comparing to other CNS and PNS tissue. What is intriguing is that there is almost no immunoreactivity of ZIKV NS2B in the thalamus and subventricular zone. What makes frontal cortex and hippocampus more susceptible to ZIKV infection? NS4A/B inhibit Akt-mTOR signaling in human fetal neural stem cells (Liang Q et al *Cell Stem Cell* 2016). Is Akt-mTOR or other related signaling pathways responsible for the preference in infection?

-- As noted above, experiments in which immunocompromised (IFN receptor deficient) mice were infected with ZIKV revealed that the pattern of brain regional distribution of viral load was lost, i.e., all brain structures investigated exhibited similarly high viral loads. This suggests that the ability to mount an effective IFN-mediated immune response may differ between brain regions/structures and may, at least in part, explain the different vulnerabilities of distinct brain regions to ZIKV infection. Nonetheless, we acknowledge that additional mechanisms may also be

implicated, including a possible differential impact of ZIKV on the mTOR (as suggested by the reviewer) or other neuronal signaling pathways, and we feel that a more detailed investigation of this issue should be pursued in future studies.

4) “The authors showed that ZIKV infection targeted neurons but not astrocytes and microglia. Do the infected neurons trigger inflammation and microgliosis? What is the possible cause of inflammation and microgliosis?”

-- Our results show that ZIKV infects neurons, but how these cells respond to infection is still poorly understood. It has been demonstrated that ZIKV activates Toll-like receptor 3 (TLR3) in neurons (Dang et al., 2016, Cell Stem Cell). It has also been reported that ZIKV infection induces phosphorylated TANK-binding kinase 1 (pTBK1) mobilization to the mitochondrial membrane (Onorati et al., 2016, Cell Rep.), suggesting that ZIKV triggers mitochondrial antiviral signaling (MAVS) and NFkB activation. This is also supported by a recent report showing that ZIKV infection triggers synthase/stimulator of IFN genes (STING-) and autophagy-dependent NFkB activation that is protective in a Drosophila model of ZIKV brain infection (Liu et al., 2018, Cell Host Microbe).

Although the mechanisms linking neuronal infection and cytokine production are still a matter of debate, these studies suggest that neurons are both a target of ZIKV infection and a potential source of inflammation. In line with this notion, it was recently shown that ZIKV infection of cultured murine neurons triggers production of the pro-inflammatory cytokines, TNF- α and IL-18 (Olmo et al., 2017, Front Immunol.). Despite the modest increase in levels of these cytokines, this was sufficient to facilitate NMDAR-dependent neuronal death. Increased neuronal death could represent an amplifying loop mechanism to increase inflammatory signals and microgliosis. Moreover, it has also been demonstrated that ZIKV infection triggers the unfolded protein response (UPR) in neurons (Gladwyn-Ng et al., 2017, Nat Neurosci.), a cellular pathway associated with microgliosis and neuroinflammation in several models of neuropathologies (review by Sprenkle et al., 2017, Mol Neurodegener.). A few sentences have been added to Discussion in the revised manuscript to highlight these previous findings (page 20, li 416 and page 21, li 417-19).

We found that infliximab abolished microgliosis in ZIKV-infected mice, suggesting that TNF- α is upstream of microglial activation in response to ZIKV infection. Because we found no evidence of astrogliosis in ZIKV-infected mice, we feel our results are consistent with the hypothesis that neurons produce pro-inflammatory cytokines as part of the response to ZIKV infection, perhaps through mechanisms that involve membrane or cytoplasmic receptors (TLR3 or RIG-I/MAVS, respectively). Such cytokines could then trigger inflammation and microglia activation, either directly or indirectly by facilitating neuronal cell death. A more detailed investigation of the interplay between neurons and microglia in the context of ZIKV infection appears warranted in follow-up studies.

5) “Two puzzling results from the behavioral tests: 1) despite similar concentration of virus as well as NS2B expression in the striatum and hippocampus, the infected mice did not exhibit locomotor and exploration deficits; 2) despite similar concentration of virus as well as NS2B expression in the frontal cortex and hippocampus, the infected mice did not exhibit anxiety deficits. It would be helpful to show whether synapse damage, brain inflammation and microgliosis occur in the striatum and frontal cortex as well.”

-- We thank the reviewer for raising these important points, and for the opportunity to perform a more detailed analysis of ZIKV-induced pathology in different brain regions.

As mentioned above (response to point #1), we carried out additional experiments to determine whether ZIKV infection was associated with motor deficits. Although our previous open field analysis had failed to reveal motor deficits (Suppl. Fig. 3a,b), we indeed found that ZIKV-infected mice showed impaired performance in the rotarod task at 5 dpi, approximately coinciding with the peak in viral replication (Suppl. Fig. 8h). Mice recovered normal motor function when viral replication was no longer active (30 dpi; Suppl. Fig. 8i). Interestingly, Iba-1 immunostaining was elevated in the striatum of ZIKV-infected mice at 5 dpi (Suppl. Fig. 8a,b,g), and returned to basal levels at 30 dpi (Suppl. Fig. 8c,d,g). Results suggest that increased striatal microgliosis is associated with poor motor performance in ZIKV-infected mice.

We also assessed microgliosis in the frontal cortex and found a non-statistically significant increase in Iba-1 immunostaining at 6 days following ZIKV infection (Suppl. Fig. 7d-f).

We further note there was a trend (which, however, did not reach statistical significance) of decrease in synaptophysin immunoreactive puncta in the striatum and frontal cortex of ZIKV-infected mice (Suppl. Fig. 4m-o).

6) "In the novel object recognition test, the variability in exploration time is much larger at 30 dpi in comparison to 1 dpi and 14 dpi. A paired t-test will be helpful to better illustrate the exploration of familiar and novel object for each mouse at different time points."

-- We would first like to clarify that independent groups of mice were tested in the NOR paradigm at 1, 14, 30 or 60 dpi. To make this clear to the reader, we have inserted a sentence stating this fact in the corresponding Figure Legend. Some of the variability in exploration times may thus be due to individual differences in animals belonging to different groups. With respect to the suggestion to use a paired t-test to compare exploration times for each mouse, we note that, due to the intrinsic nature of the NOR test, the fractions of time spent exploring the novel and familiar objects are not independent variables (i.e, they add up to 100%). Thus, a paired t-test would not appear indicated in this case. There are two main ways to analyze NOR data in the published literature: (i) by comparing the fraction of time spent exploring the novel object with the chance value of 50%, using a one-sample t-test, as we have done in the current and in previous studies (e.g., Figueiredo et al., 2013, J Neurosci.; Lourenco et al., 2013, Cell Metab.; Ledo et al., 2013, Mol Psychiatry), or (ii) performing ANOVA comparing the fraction of exploration times of the novel object among distinct experimental groups. In the current study, we have used the former approach as a well-accepted procedure in the literature.

7) "What is the virus concentration or NS2B signal in other cortical areas such as motor, somatosensory or entorhinal cortices? This result may provide some evidences for memory but not motor or sensory deficits."

-- Following the reviewer's suggestion, we have performed additional experiments to assess viral distribution in the entorhinal, somatosensory and motor cortices. Results show that viral load is ~ 100-fold lower in the motor cortex and ~ 1,000-fold lower in somatosensory and entorhinal cortices than in the hippocampus (Suppl. Fig. 1b).

Prompted by the reviewer's comment, we have further evaluated performance of mice in the von Frey, cold (acetone) and heat (Hargreaves) sensitivity tests at different times following infection to determine whether ZIKV infection induced sensory deficits. We found no differences in performance between mock- and ZIKV-infected mice in these tests (Suppl. Fig. 8j-l).

8) "The authors showed a loss of pre-synaptic terminals in CA1 and dentate gyrus of ZIKV-infected adult mice. Majority of synaptic input into CA1 and dentate gyrus are from entorhinal cortex (located in the medial temporal lobe), with some local input from CA3. Therefore, part of the presynaptic loss in CA1 and dentate gyrus is not from hippocampus but cortex. In order to clarify which synapses go wrong in the hippocampus, it would be helpful to show pre-synaptic (Synaptophysin) as well as postsynaptic (Homer1) puncta using Synaptophysin and Homer1 immunostaining in CA1 and dentate gyrus. It would be even better to show the data in CA3 as well.

-- We thank the reviewer for the opportunity to examine in more detail the synaptic damage induced by ZIKV infection in adult mice. As requested, we have performed double immunofluorescence labeling of synapses using pre- and post-synaptic markers (synaptophysin and Homer1, respectively) followed by confocal imaging in brain sections from ZIKV-infected mice at 6 and 60 dpi (new Fig. 3g-p). We found that, at the peak of viral replication (6 dpi), the CA3 hippocampal subregion of ZIKV-infected mice presented significantly decreased levels of pre-synaptic (new Fig. 3g-i) but not post-synaptic (new Fig. 3g,h,j) puncta, resulting in lower co-localized synaptic puncta (new Fig. 3g,h,k) compared to mock-infected mice. Interestingly, synapse damage induced by ZIKV was reversible, as pre-synaptic and col-localized synaptic puncta returned to control, mock-infected levels at 60 dpi (Fig. 3 l-p). These findings corroborate results from Western blot analyses of mouse hippocampal tissue for pre-synaptic (synaptophysin) and post-synaptic (PSD-95) proteins (Suppl. Fig. 4f-g in the revised manuscript).

9) “Memory deficits lasted for at least 30 days after infection and recovered at 60 dpi. However, the authors provided data for synapse damage, brain inflammation and microgliosis only at 6 dpi. To demonstrate a close correlation between the memory deficits and synaptic/inflammation changes, the authors are suggested to show the same data at 30 and 60 dpi, respectively.”

-- Following the reviewer’s suggestion, we investigated whether synapse damage and microgliosis persisted at longer time points after infection. We found that Iba-1 immunostaining in the hippocampus, which we had reported to be elevated at 6 dpi, remained increased at 30 dpi and returned to control levels at 60 dpi (new Fig. 4c). Moreover, evaluation of colocalization of Homer-1 and synaptophysin immunostaining showed that synapse density was reduced at 6 dpi and had returned to normal levels at 60 dpi (Fig. 3g-p).

10) “The authors showed the rescue of memory deficits using Infliximab, a TNF- α antibody. It is not clear whether the inflammation directly leads to memory deficits or indirectly causes memory deficits through synapse damage. It will be helpful to show the immune data of synaptophysin as well as Homer1 in the Infliximab-rescued mice.

-- Following the reviewer’s suggestion, we have performed double-immunofluorescence labeling of synaptophysin/Homer1 to assess synapse density in infliximab-treated mice. Results showed that treatment with infliximab blocked the decreases in synaptophysin immunoreactivity and in co-localized synaptophysin/Homer1 puncta in the hippocampi of ZIKV-infected mice (Fig. 5c-h). Results thus suggest that inflammation leads to synapse damage, which likely explains memory deficits in ZIKV-infected mice.

Minor comments:

“1. In page 6, the authors claim that Fig. 1j showed staining of pyramidal and granule cells in the hippocampus. How did the authors identify that the cells are pyramidal and granule cells?”

-- We apologize for the misleading description of results. Immunostaining for ZIKV

NS2B was performed in coronal brain sections from ZIKV- or mock-infected mice, corresponding to between -1.64 and -1.68 mm from Bregma, and between +2.16 and +2.12 mm interaural. Reviewer Fig. 1 illustrates one such coronal brain section including the dorsal hippocampus of mice. Thus, the correct form of the sentence, as now corrected in the revised manuscript, should read “Immunostaining for ZIKV NS2B protein in coronal brain sections from mice infected by i.c.v. route (Fig. 2f) revealed robust staining in the hippocampal pyramidal cell layer (Fig. 2g-i) and in the granular layer of the dentate gyrus (Fig. 2g,j).”.

Reviewer Fig. 1. Coronal section of the mouse brain showing the dorsal hippocampus. (Py) Pyramidal cell layer of the hippocampus in the CA1 and CA2 subregions. (GrDG) Granular layer of dentate gyrus. From MLB Mouse Brain Atlas (http://www.mbl.org/atlas170/large_label/20.jpg).

“2. In page 6-7, what is the brain region for Fig. 1 t-j’?”

-- Original Fig. 1t-j’ shows CA1 hippocampal sub-region of adult ZIKV-infected mice. These panels are shown in new Fig. 2p-f’. To clarify this point, we have included brain region information in the legend to this figure in the revised manuscript.

“3. In page 10, Suppl. Fig. 3h-k should be Suppl. Fig. 2h-k.”

-- We apologize for the typo. We have corrected the Figure citation in the revised manuscript.

“4. In page 12, n = 7 - 10 mice/group for Fig. 2d. Please specify the exact number of mice in each group.”

-- We note that the exact number of animals used in each experiment is shown by the number of symbols in the scatter plots used throughout the manuscript. Nonetheless, as suggested by the reviewer and for additional clarity, we have included the exact number of mice per group in Figure Legends.

"5. In page 19, ...(triplly deficient in interferon regulatory factor)25. This reference should be reference 9."

-- We apologize for the mistake, which has been corrected in the revised version.

"6. In page 40-41, references 7 and 12 are the same."

-- Once again, we thank the reviewer for the detailed examination of our manuscript. The list of references has been verified and corrected in the revised manuscript.

Reviewers' comments:

Reviewer #1 (Remarks to the Author):

The authors of this paper have attempted to respond to reviewers' concerns primarily by adding additional experiments. Unfortunately, the new results increase the amount of descriptive and correlative information with regard to their model but do not address the weaknesses in their interpretations of their own data or the requested further evaluation of mechanisms of pathogenesis. Thus, the revised paper continues to fall short of the bar for publication in Nature Communications. Most importantly, the authors continue to conclude that they have developed a model of post-infectious memory dysfunction after ZIKV infection, when it is more likely that alterations in behaviors are due to sickness behaviors during acute encephalitis. These occur in the setting of most viral infections in the CNS, not only ZIKV, and the notion that their behavioral findings are due to acute effects is supported by the resolution of all dysfunction and pathology by day 60 post-infection. If the authors wanted to model post-infectious cognitive dysfunction, they would need to show that memory impairment and pathology persist long after viral clearance. There is also a lack of data supporting the conclusions that a "microglia/TNF/complement axis" underlies their findings, as additional data suggest that macrophages are involved in these acute processes, and that TNF levels are unchanged in the setting of minocycline treatment, which improves behaviors assessed. Minocycline reportedly decreases microglial activation, which is not evaluated in the setting of acute ZIKV encephalitis here, nor are there data demonstrating that microglia engulf synapses or even interact with them. Thus, the revised paper continues to be misleading, mostly correlative and without adding novel information regarding mechanisms of behavioral alterations in a model of acute encephalitis. Specific examples regarding response to criticisms:

1. The original manuscript was criticized for lack of thorough examination of human tissues, including identification of all NS2B+ cells, which has not been included in the revised manuscript. These studies also do not include any immunohistochemical controls, which are required given that a single cell is depicted in Figure 1. Results examining NS2B staining in murine CNS tissues are also confusing, as there is a lack of demonstration of activated microglia or astrocytes and no apparent detection of NS2B.
2. The original manuscript was criticized for concluding that a model of viral encephalitis in which memory dysfunction improves after clearance of virus is not a model of post-infectious memory impairment. It is well-established that acute viral encephalitis is associated with sickness behaviors that may decrease performance on behavioral tests and including fatigue, hypersomnia, depressed activity, decreased social interactions and inability to concentrate (Reviewed in Brain Behav Immun 50, 322-333, doi:10.1016/j.bbi.2015.07.012 2015). As the authors continue to focus their interpretations on a single brain region and demonstrate that mice improve after viral

clearance, the interpretation that ZIKV specifically impacts on hippocampus-based memory and synaptic plasticity is misleading.

3. Why do mock-infected mice exhibit alterations in behavior at all time-points? (S3a)
4. The complete recovery of synaptophysin staining at 60 days supports the notion that behavioral alterations are the results of acute encephalitis with sickness behaviors and do not model post-infectious cognitive dysfunction. Also, how do the authors explain the reversed results in the DG?
5. Synaptic plasticity is defined as the ability of synapses to strengthen or weaken over time. Electrophysiology data only includes one time-point. Thus, the use of the term “synaptic plasticity” in discussions of these data and in the title is incorrect.
6. Despite concluding in their response that infiltrating macrophages likely play a role in the pathogenesis of acute ZIKV encephalitis, this is not included in the abstract, final paragraph of the introduction and not discussed.
7. The authors also do not the direct mechanisms of TNF-mediated alterations in behavior. Indeed, the lack of effect of minocycline on TNF (Fig 5n) argues that this protein is not involved in the amelioration of behavior in the setting of minocycline treatment. It also suggests that the effects of minocycline on macrophage/microglia activation are unrelated to any effects of TNF on behavior.
8. The authors do not explore the request to determine whether the source of complement is serum. This is important because the authors conclude that microglia are the source of complement whereas viral encephalitis is often associated with increased BBB permeability.
9. The authors’ conclusion in the revised manuscript that synapse alteration occurs via a complement/microglia/TNF axis is not supported by the existing data, as there are no data demonstrating directly that microglia interact with or engulf synapses, or data supporting any direct relationships between microglial-mediated synapse loss, TNF and complement. In addition, the authors state that minocycline alters the polarization of microglia, a concept that has been largely abandoned by microglia experts. Thus, the manuscript continues to reproduce some findings in previously published work using other models of flavivirus encephalitis (Mol Neurobiol. 2017 Aug;54(6):4705-4715; Nature. 2016 Jun 23;534(7608):538-43) continues to be largely correlative, and does not provide any novel information regarding mechanisms of behavioral alterations during acute and resolving viral encephalitis, including those that might be specific for ZIKV.

Reviewer #3 (Remarks to the Author):

The authors have conducted new experiments to address my concerns. The overall quality of the manuscript has improved from the last version.

I have one last comment for the authors:

In Fig. 3 q&r, the authors measured the fEPSP after ZIKV infection. However, in the rescue experiments thereafter, namely blockage of microglial polarization or neutralization of TNF- α signaling or C1q/C3, fEPSP was not measured.

RESPONSE TO REFEREES LETTER

Reviewer #1:

General comments:

“...the authors continue to conclude that they have developed a model of post-infectious memory dysfunction after ZIKV infection, when it is more likely that alterations in behaviors are due to sickness behaviors during acute encephalitis. These occur in the setting of most viral infections in the CNS, not only ZIKV, and the notion that their behavioral findings are due to acute effects is supported by the resolution of all dysfunction and pathology by day 60 post-infection. If the authors wanted to model post-infectious cognitive dysfunction, they would need to show that memory impairment and pathology persists long after viral clearance.”

-- We thank the reviewer for the constructive criticism. We fully agree that memory deficits induced by ZIKV infection are reversible, as mice recover normal cognitive function at 60 days post infection (dpi), a timepoint when no viral RNA or brain inflammation remain. However, the point we have tried to convey is that memory impairment in mice persists for some time after infection and inflammation have been resolved. Specifically, results show that animals exhibited clear memory impairment at 30 dpi, a time point at which infection had subsided (as indicated by a brain viral load close to the detection limit) and cytokines had returned to control levels, indicating animals have recovered from acute brain inflammation. Nonetheless, we feel the term “persistent” we used in previous versions of the manuscript is subjective, and may be misleading as suggested by the Reviewer. Accordingly, we have revised the manuscript to ensure that we no longer refer to ZIKV-induced memory impairment as “persistent”.

The Reviewer was further concerned whether memory dysfunction in ZIKV-infected mice might be due to sickness behavior. This is indeed an important point, as viral infections have been shown to cause general and nonspecific behavioral symptoms, including lethargy,

immobility, sleepiness, and depressive- or anxiety-like behaviors (Hart, *Neurosci. Biobeh. Rev.*, 1988; Dantzer et al., *Nat. Rev. Neurosci.*, 2008; Blank et al., *Immunity*, 2016; Blank & Prinz, *Glia*, 2017). In the previous version of our manuscript, we had shown that mock-infused and ZIKV-infected mice traveled comparable distances and spent similar amounts of time at the center of an open field arena (Suppl. Fig. 3a, f), indicating that locomotor/exploratory and anxiety-like behaviors were not affected by ZIKV infection. These observations are further supported by the fact that no difference was found between mock- and ZIKV-infected mice between the amount of time spent in the open arms of the elevated plus maze (Suppl. Fig. 3h). In addition, both groups showed comparable total exploration times towards the objects used in both training and test sessions of the novel object recognition (NOR) test (Suppl. Fig. 4a-h), indicating that infection was not associated with lethargy or reduced exploratory interest.

To examine in more detail the possibility that ZIKV infection could be associated to sickness behavior, mock-infused and ZIKV-infected mice were evaluated during the acute phase of infection (4 dpi) in the tail suspension (TST) and sucrose splash tests (SST), tasks designed to evaluate behavioral despair and/or depressive-like behavior and anhedonia, common features of sickness behavior (Steru et al., *Psychopharm.*, 1985; Yalcin et al., *Beh. Brain Res.*, 2008; Dantzer et al., *Nat. Rev. Neurosci.*, 2008; O'Connor et al., *J. Immunol.*, 2010). We found no differences between mock-infused and ZIKV-infected animals in either immobility time in the TST (new Suppl. Fig. 3i) or in grooming behavior in the SST (new Suppl. Fig. 3j-k). We further expanded behavioral evaluation of mock- and ZIKV-infused mice by allowing animals to explore the open field arena for 30 min. Again, we found no differences between mock-infused and ZIKV-infected animals in distance travelled, number of body rotations, rearings or time spent at the center of the arena (new Suppl. Fig. 3 c-e, g). Moreover, both mock- and ZIKV-infected mice showed the expected locomotor habituation

in the open field test (new Suppl. Fig. 3b). Altogether, results thus indicate that ZIKV infection did not instigate depressive-like/sickness behavior in mice.

We further performed additional experiments to determine whether memory impairment triggered by ZIKV might be a nonspecific consequence of brain inflammation, and could be equally triggered by infection by any other arbovirus or even by inactivated ZIKV. Mice were first infused with 10^5 PFU UV-inactivated ZIKV (iZIKV) and evaluated in the NOR task. As shown in new Suppl. Fig. 4k-o, iZIKV did not impact memory in mice. Second, under the same conditions used for ZIKV infection, mice were infected i.c.v. with 10^5 PFU Mayaro virus (MAYV), another emergent arbovirus. At the peak of replication of MAYV in the mouse brain (1 dpi) as well as 7 and 14 dpi, mice were assessed in the NOR task. As shown in new Suppl. Figs. 4 p-u, MAYV infection had no impact on memory in mice.

Collectively, results indicate that memory impairment is not caused by brain infection by any arbovirus and that impairment triggered by ZIKV is not caused by induction of a general sickness behavior. We have included these new findings and a brief discussion on sickness behavior in the revised manuscript.

“There is also a lack of data supporting the conclusions that a “ microglia/TNF/complement axis” underlies their findings, as additional data suggest that macrophages are involved in these acute processes, and that TNF levels are unchanged in the setting of minocycline treatment, which improves behaviors assessed. Minocycline reportedly decreases microglial activation, which is not evaluated in the setting of acute ZIKV encephalitis here, nor are there data demonstrating that microglia engulf synapses or even interact with them.”

-- We thank the reviewer for the opportunity to further discuss the involvement of microglia, TNF- α and complement in ZIKV-induced cognitive impairment. To address this issue, we first examined the effectiveness of minocycline treatment in preventing microglial activation

in the context of ZIKV infection. Treatment of ZIKV-infected mice with minocycline did not affect the number of Iba-1-positive cells in the hippocampus (new Fig. 5d-g), but blocked the increase in Feret's diameter (a morphological index of microglial activation) induced by ZIKV (new Fig. 5h). This indicates that minocycline blocks the switch of microglia to an amoeboid morphology characteristic of a phagocytic activated state upon ZIKV infection (Fig. 4e-j). Importantly, minocycline treatment mitigated memory impairment (Fig. 5j) and synapse dysfunction (new Fig. 5k-n) induced by ZIKV infection, supporting the notion that microglia activation is linked to synaptic and memory deficits induced by ZIKV.

Interestingly, minocycline did not interfere with brain TNF- α expression (Fig. 5i) in ZIKV-infected mice. This might initially appear counterintuitive as microglia are known to produce and secrete TNF- α upon activation. It is well known, however, that TNF- α is produced by a variety of immune cells, including T and B lymphocytes, natural killer cells, and monocytes (Turner and Feldmann, *Biochem. Biophys. Res. Commun.*, 1988; English et al., *J. Biol. Chem.*, 1991; Andersson et al., *J. Immunol. Methods*, 1989; Santis et al., *Eur. J. Immunol.*, 1992; Wang et al., *J. Leukoc. Biol.*, 2012; Satoh et al., *Int. J. Cardiol.*, 2006). Moreover, non-immune cells, including neurons, also produce TNF- α (Nelson et al., *J. Neurosci.*, 2013; Janelins, *Am. J. Pathol.*, 2008; Cowan et al., *J. Virol.*, 1997). Specifically, a recent study showed increased levels of TNF- α and IL-1 β in the supernatant of ZIKV-infected murine neuronal primary cultures (Olmo et al., *Front. Immunol.*, 2017). Data shown in the previous version of our manuscript indicated that ZIKV infection induces not only microgliosis, but also brain infiltration by B and T lymphocytes and, possibly, macrophages, which could be sources of increased brain TNF- α in infected mice. Results further show that increased brain TNF- α is involved in ZIKV-induced microgliosis, synapse damage/dysfunction and memory deficits (Fig. 6, including new results on LTP discussed below). Results thus suggest that TNF- α production is upstream of microgliosis in the context

of ZIKV infection but is not sufficient to cause cognitive impairment in the absence of microglial activation. A brief discussion in this regard has been added to the revised manuscript (page 16, lines 359-63; page 23, lines 505-16).

Finally, to better characterize the involvement of microglia in ZIKV-induced synapse damage and loss, we performed three-dimensional image reconstructions of Iba-1-positive cells in the hippocampi of ZIKV-infected mice. Colocalization between presynaptic marker protein, synaptophysin, and Iba-1 revealed increased numbers of presynaptic terminals inside microglial cells in the hippocampi of ZIKV-infected mice compared to mock-infected animals (new Fig. 5a-c). In conjunction with data shown in the previous version of the manuscript, including synapse preservation in animals treated with infliximab, anti-C1q or anti-C3 antibodies (new Fig. 6f-k and 7g-m), current results support a link between TNF- α , complement system proteins, microglia-mediated synapse loss and memory impairment in ZIKV-infected mice.

Specific points:

1. *“The original manuscript was criticized for lack of thorough examination of human tissues, including identification of all NS2B+ cells, which has not been included in the revised manuscript. These studies also do not include any immunohistochemical controls, which are required given that a single cell is depicted in Figure 1.”*

-- We thank the reviewer for these pertinent comments. Regarding immunohistochemical controls, we note that usual controls have been performed for specificity of the secondary antibodies used (i.e., in the absence of the corresponding primary antibodies), as well as for specificity of the anti-NS2B antibody (in mock-infected slices). As expected, in both cases (absence of primary anti-NS2B antibody and mock-infected tissue) no cell-specific labeling was detected.

In order to better characterize the cell type(s) preferentially infected by ZIKV in human tissue, we performed double immunohistochemistry to detect ZIKV polyprotein (NS2B) 24 h after infection (Fig. 1d, g, j) in neurons (NeuN), astrocytes (GFAP) or microglial cells (F4/80). NS2B immunostaining co-localized with cells immunoreactive for NeuN (a marker of mature neurons) (Fig. 1c-e), but not for GFAP (Fig. 1f-h) or F4/80 (Fig. 1i-k), indicating that neurons are targeted by ZIKV in the mature human brain tissue.

“Results examining NS2B staining in murine CNS tissues are also confusing, as there is a lack of demonstration of activated microglia or astrocytes and no apparent detection of NS2B.”

-- We appreciate the reviewer's concern regarding the representative images of NS2B+ cells in the mouse brain used in Fig. 2. First, we would like to bring to the reviewer's attention data from the original version of our manuscript, where we showed no increase in immunoreactivity for GFAP in the hippocampus of ZIKV-infected mice compared to mock-infected animals (Suppl. Fig. 10). Therefore, the fact that representative images in Fig. 2 do not show astrogliosis is consistent with our previous results.

Concerning the double immunostaining for F4/80 and NS2B, we agree that the images included in the first revised version of the manuscript did not adequately represent activated/amoeboid microglial cells. We do, however, maintain that the images we had used to illustrate this figure show that NS2B labeling does not coincide with F4/80 labeling. We further note that ZIKV-induced microgliosis and microglial activation in infected mice is well characterized by results shown in Fig. 4 and Suppl. Fig. 8-9 of our study. In any case, to address the reviewer's concern, the representative images for this experiment have been replaced by new ones in the revised version of the manuscript (new Fig. 2 s-x).

2. *“The original manuscript was criticized for concluding that a model of viral encephalitis in which memory dysfunction improves after clearance of virus is not a model of post-infectious memory impairment. It is well-established that acute viral encephalitis is associated with sickness behaviors that may decrease performance on behavioral tests and including fatigue, hypersomnia, depressed activity, decreased social interactions and inability to concentrate (Reviewed in Brain Behav Immun 50, 322-333, doi:10.1016/j.bbi.2015.07.012 2015).”*

-- The issue of whether ZIKV infection induces sickness behavior which could be responsible for the observed impairment in memory tests has been addressed in response to the Reviewer’s first general comment (please see above).

“As the authors continue to focus their interpretations on a single brain region and demonstrate that mice improve after viral clearance, the interpretation that ZIKV specifically impacts on hippocampus-based memory and synaptic plasticity is misleading.”

-- Our conclusion that ZIKV preferentially targets memory-related brain regions is based on analysis of viral distribution in the brain following infection, which revealed higher viral titers in hippocampus, frontal cortex and striatum of ZIKV-infected mice. Thus, we respectfully disagree with the interpretation that we have “focused our interpretation on a single brain region”. Instead, our decision to focus our investigation on memory-related regions was strictly based on results from viral distribution analysis. We further note that, in addition to memory, we have evaluated the impact of ZIKV infection on a number of other brain functions, including pain sensitivity (to mechanical and thermal stimuli), motor activity, anxiety- and depressive-like behavior. Consistent with striatum targeting by ZIKV, we further found that motor activity, but not the other functions investigated, was impaired in ZIKV-infected mice.

3. *“Why do mock-infected mice exhibit alterations in behavior at all time-points? (S3a)”*

-- Graphs shown in Suppl. Fig. 3a, b represent data from the same groups of mice that were assessed in the open field arena at sequential time points after infection. Habituation in an open-field environment, expressed as a decrease in exploratory activity in subsequent assessments, is a well-known behavioral outcome associated with contextual learning (Crusio & Schwegler, Beh. Brain. Res., 1987; Corey, Neurosc. Biobeh. Rev., 1978). As expected, both mock-infused and ZIKV-infected mice showed decreased locomotor activity in subsequent sessions, reflecting habituation to the open field (Suppl. Fig. 3a). To further clarify this point, we have added a couple of sentences in the revised version of the manuscript to discuss these observations (page 10, lines 212-14).

4. *“The complete recovery of synaptophysin staining at 60 days supports the notion that behavioral alterations are the results of acute encephalitis with sickness behaviors and do not model post-infectious cognitive dysfunction.”*

-- As discussed above (please see our reply to the Reviewer’s first general comment, above), ZIKV-infected animals exhibited neither decreased exploratory/locomotor activities (Suppl. Fig. 3a-e) nor depressive/anxiety-like behaviors, common features of sickness behavior in mice (Suppl. Fig. 3 f-k). These observations are discussed above and in the new version of the manuscript (page 10, lines 208-29; page 21, lines 465-67). Nonetheless, taking into account the Reviewer’s concern, we have carefully revised the manuscript so as to no longer refer to ZIKV-induced cognitive impairment as “persistent”.

“Also, how do the authors explain the reversed results in the DG?”

-- We believe the reviewer is referring to the fact that the decrease in synaptic puncta (co-localized synaptophysin/Homer-1 labeling) in the CA3 region is mostly explained by the decrease in synaptophysin immunoreactivity, whereas in the DG decreased Homer-1

immunoreactivity is mainly responsible for the decrease in co-localized synaptic puncta. This is indeed an interesting point, and we thank the reviewer for bringing it up. The DG is the first hippocampal structure to receive cortical input, and neuronal cell bodies in the DG (mossy cells) send excitatory projections (mossy fibers) to the CA3 region. Our finding that post-synaptic Homer-1 is mostly decreased in the DG, while pre-synaptic synaptophysin is mostly decreased in the CA3 region suggests that mossy cells are particularly susceptible to ZIKV infection. We further note this is precisely why we have performed LTP measurements in the mossy fiber circuit in the current study, rather than the more usually examined Schaeffer collaterals (CA3-CA1) pathway. Although we feel a more detailed investigation of this hypothesis would be beyond the scope of this study.

5. *“Synaptic plasticity is defined as the ability of synapses to strengthen or weaken over time. Electrophysiology data only includes one time-point. Thus, the use of the term “synaptic plasticity” in discussions of these data and in the title is incorrect.”*

-- We are well aware of the fact that LTP is not synonymous with synaptic plasticity. In fact, synaptic plasticity encompasses both much shorter process, such as those involved in paired-pulse facilitation (PPF), and the very long (years, decades) processes underlying human memory. Our electrophysiological data clearly show that ZIKV infection disrupted PPF, post-tetanic potentiation and LTP, thus indicating an impact on multiple processes involved in synaptic plasticity. We further note that LTP is widely recognized as a form of synaptic plasticity that is central for memory formation, and it is considered a reliable measurement of the efficiency of synaptic function (e.g., Whitlock et al., Science, 2009; Ortiz et al., J Neurosci, 2010; Engert and Bonhoeffer, Nature 1999; Matsuzaki et al, Nature, 2004). Thus, we have used the term “synaptic plasticity” (as do numerous other papers in the memory field) to indicate that ZIKV impacts synaptic function by inhibiting synaptic strengthening following high-frequency stimulation, a form of plasticity that is germane to memory

processes. Nonetheless, to address the reviewer's concern, we have replaced the term "synaptic plasticity" by "synaptic function" in the title and throughout the text.

6. *"Despite concluding in their response that infiltrating macrophages likely play a role in the pathogenesis of acute ZIKV encephalitis, this is not included in the abstract, final paragraph of the introduction and not discussed."*

-- We thank the reviewer for noting this omission. We have now explicitly mentioned infiltrating macrophages in the revised manuscript.

7. *"The authors also do not the direct mechanisms of TNF-mediated alterations in behavior. Indeed, the lack of effect of minocycline on TNF (Fig 5n) argues that this protein is not involved in the amelioration of behavior in the setting of minocycline treatment. It also suggests that the effects of minocycline on macrophage/microglia activation are unrelated to any effects of TNF on behavior."*

-- We thank the reviewer for the opportunity to further clarify this important point. We first note that minocycline decreased conversion of microglia to an amoeboid morphology (new Fig. 5d-f, h) and mitigated cognitive impairment (Fig. 5j) and synapse damage (new Fig. 5k-n) induced by ZIKV infection, even though it did not interfere with brain expression of TNF- α (Fig. 5i). It is well known that TNF- α is produced by a variety of immune cells, including T and B lymphocytes, natural killer cells, and monocytes (Turner and Feldmann, *Biochem. Biophys. Res. Commun.*, 1988; English et al., *J. Biol. Chem.*, 1991; Andersson et al., *J. Immunol. Methods*, 1989; Santis et al., *Eur. J. Immunol.*, 1992; Wang et al., *J. Leukoc. Biol.*, 2012; Satoh et al., *Int. J. Cardiol.*, 2006). Moreover, non-immune cells, including neurons, also produce TNF- α (Nelson et al., *J. Neurosci.*, 2013; Janelins, *Am J Pathol.*, 2008; Cowan et al., *J. Virol.*, 1997). In addition, a recent study showed increased levels of TNF- α and IL-1 β in the supernatant of ZIKV-infected murine neuronal primary cultures compared to mock-infected cultures (Olmo et al., *Front. Immunol.*, 2017). Our results show that infection by

ZIKV induces not only microgliosis, but also brain infiltration by T and B lymphocytes, as well as likely macrophages. Results thus suggest that TNF- α production is upstream of microglial activation (as supported by the fact that infliximab partially blocks microgliosis in ZIKV-infected mice; Fig. 6a-e), and show that increased brain TNF- α production is involved in ZIKV-induced memory deficits, but is not sufficient to cause cognitive impairment in the absence of microglial activation (Fig. 5i-j). A brief discussion in this regard has been added to the revised manuscript (page 16 359-63; page 23, lines 505-516).

8. *“The authors do not explore the request to determine whether the source of complement is serum. This is important because the authors conclude that microglia are the source of complement whereas viral encephalitis is often associated with increased BBB permeability.”*

-- We note that we have analyzed the expression (mRNA) of C1q and C3 in the brains of ZIKV-infected mice. While results clearly indicate there is increased brain production of these factors in infected animals, they do not rule out the possibility that additional complement proteins could come from the plasma. Accordingly, as in the previous version of the manuscript, we discuss the possible sources of C1q and C3, as transcribed below:

“Microglia were recently shown to play a central role in the pruning of synapses tagged by complement system proteins in a mouse model of WNV infection. Significantly, we found that treatments with infliximab or minocycline prevented memory impairment in ZIKV-infected mice. Brains of ZIKV-infected mice showed increased expression of complement system proteins, C1q and C3. Further implicating TNF- α , microglia and complement activation in ZIKV-instigated synapse loss and memory deficits, inhibition of TNF- α signaling prevented synapse damage, microglial activation and attenuated the increase in brain levels of complement system protein C3 in ZIKV-infected mice. We note, nonetheless, that complement in the CNS in ZIKV infected mice may originate from disruption of BBB integrity and infiltration of peripheral immune cells, as well as from activation of resident

microglial cells, a possibility that is supported by our finding that both central (i.c.v.) and peripheral (i.p.) treatments with infliximab resulted in decreased brain expression of C3. On the other hand, the lack of effect of infliximab on brain C1q expression in ZIKV-infected mice may be related to the capacity of neurons to express C1q. Importantly, we show that i.c.v. infusion of either anti-C1q or anti-C3 antibodies preserves synapses and prevents memory impairment in ZIKV-infected mice, indicating a central role of complement activation in ZIKV-induced synapse damage and memory impact.”

9. “The authors’ conclusion in the revised manuscript that synapse alteration occurs via a complement/microglia/TNF axis is not supported by the existing data, as there are no data demonstrating directly that microglia interact with or engulf synapses, or data supporting any direct relationships between microglial-mediated synapse loss, TNF and complement. In addition, the authors state that minocycline alters the polarization of microglia, a concept that has been largely abandoned by microglia experts.”

-- As suggested by the reviewer, to better characterize the involvement of microglia in ZIKV-induced synapse damage and elimination, we have performed three-dimensional reconstructions from confocal sections of Iba-1 positive cells from the hippocampi of ZIKV- or mock-infected mice. This revealed increased numbers of synaptophysin-labeled presynaptic terminals inside Iba-1-positive cells in ZIKV-infected mice (new Fig. 5a-c), similar to previous studies that have shown synapse engulfment and pruning by microglia (Hong et al., Science 2016; Vasek et al., Nature, 2017; Schafer et al., Neuron, 2012). In association with data shown in the previous version of the manuscript, including synapse preservation in animals treated with infliximab, anti-C1q or anti-C3 antibodies (new Fig. 6f-k and 7g-m), we feel the new imaging results provide compelling evidence to support a direct relationship between microglial activation and synapse loss in the context of ZIKV infection.

Concerning the statement that minocycline inhibits microglial polarization, we fully agree with the reviewer that the concept of microglial polarization strictly between two states (M1 and M2) has been progressively replaced by the view that microglia exist in a spectrum of activation states depending on the type of stimulus provided. In addition, we agree that, despite the well documented ability of minocycline to limit neuroinflammation, the precise mechanism of action of minocycline is still under discussion. Nonetheless, minocycline is an important and widely used tool to investigate the involvement of microglial cells in the pathogenesis of many diseases (Frost et al., Cell Death Dis, 2019; Ahmed et al., Sci Rep, 2017; Quick et al., J Virol, 2017).

We have examined the efficiency of minocycline in blocking microglial activation under our experimental conditions by investigating its effect on microglial morphological conversion from a more ramified (surveilling) to a more amoeboid (phagocytic) morphology, which is well known to follow microglial activation. We found that treatment of ZIKV-infected animals with minocycline caused no change in the number of hippocampal Iba-1 positive cells (new Fig. 5g), but blocked the increase in Feret's diameter induced by ZIKV (new Fig. 5h). This result shows that minocycline blocks the switch of microglial cells to a predominantly amoeboid morphology after viral infection (Fig. 4i). In order to address the reviewer's concern, we have replaced the term "polarization" to "activation" throughout the manuscript.

Reviewer #3:

"In Fig. 3 q&r, the authors measured the fEPSP after ZIKV infection. However, in the rescue experiments thereafter, namely blockage of microglial polarization or neutralization of TNF- α signaling or C1q/C3, fEPSP was not measured."

-- We thank the Reviewer for the opportunity to extend our study of the mechanisms underlying the rescue of hippocampal synaptic function by minocycline and infliximab. As

suggested, we have performed fEPSP recordings in hippocampal slices from ZIKV-infected mice that were treated with infliximab or minocycline. As shown in new Figs. 5 k-n and 6m-p, both treatments prevented the inhibition of long-term potentiation (LTP) in the mossy fibers of ZIKV-infected mice.

REVIEWERS' COMMENTS:

Reviewer #1 (Remarks to the Author):

The authors have addressed all my concerns adequately. The manuscript is improved in its specificity in language and results.

Reviewer #3 (Remarks to the Author):

The authors have answered my question. I don't have any further comment/concern.